# *SOD1* is a synthetic-lethal target in *PPM1D*-mutant leukemia cells

Linda Zhang[1,2,3,4,5], Joanne I Hsu[1,2,3], Etienne D Braekeleer[6], Chun-Wei Chen[3,4,5,7], Tajhal D Patel[8], Alejandra G Martell[4], Anna G Guzman[4], Katharina Wohlan[4], Sarah M Waldvogel[2,3,4,5,9], Hidetaka Uryu[10], Ayala Tovy[3,4,5], Elsa Callen[11], Rebecca L Murdaugh[3,4,5,12], Rosemary Richard[3,4,5,12], Sandra Jansen[13], Lisenka Vissers[13], Bert BA de Vries[13], Andre Nussenzweig[11], Shixia Huang[4,14], Cristian Coarfa[4], Jamie Anastas[3,4,5,12], Koichi Takahashi[10,15], George Vassiliou[6], Margaret A Goodell[3,4,5]*

[1]Translational Biology and Molecular Medicine Graduate Program, Baylor College of Medicine, Houston, United States; [2]Medical Scientist Training Program, Baylor College of Medicine, Houston, United States; [3]Stem Cells and Regenerative Medicine Center, Baylor College of Medicine, Houston, United States; [4]Department of Molecular and Cellular Biology, Baylor College of Medicine, Houston, United States; [5]Center for Cell and Gene Therapy, Houston, United States; [6]Department of Haematology, Wellcome-MRC Cambridge Stem Cell Institute, University of Cambridge, Cambridge, United Kingdom; [7]Integrated Molecular and Biomedical Sciences Graduate Program, Baylor College of Medicine, Houston, United States; [8]Texas Children's Hospital Department of Hematology/Oncology, Baylor College of Medicine, Houston, United States; [9]Cancer and Cell Biology Graduate Program, Baylor College of Medicine, Houston, United States; [10]Department of Leukemia, The University of Texas MD Anderson Cancer Center, Houston, United States; [11]Laboratory of Genome Integrity, National Cancer Institute, National Institute of Health, Bethesda, United States; [12]Department of Neurosurgery, Baylor College of Medicine, Houston, United States; [13]Donders Centre for Neuroscience, Radboud University Medical Center, Nijmegen, Netherlands; [14]Department of Education, Innovation and Technology, Advanced Technology Cores, University of Texas, Houston, United States; [15]Department of Genome Medicine, The University of Texas MD Anderson Cancer Center, Houston, United States

*For correspondence: goodell@bcm.edu

Competing interest: The authors declare that no competing interests exist.

**Abstract** The DNA damage response is critical for maintaining genome integrity and is commonly disrupted in the development of cancer. PPM1D (protein phosphatase Mg$^{2+}$/Mn$^{2+}$-dependent 1D) is a master negative regulator of the response; gain-of-function mutations and amplifications of *PPM1D* are found across several human cancers making it a relevant pharmacological target. Here, we used CRISPR/Cas9 screening to identify synthetic-lethal dependencies of *PPM1D*, uncovering superoxide dismutase-1 (SOD1) as a potential target for *PPM1D*-mutant cells. We revealed a dysregulated redox landscape characterized by elevated levels of reactive oxygen species and a compromised response to oxidative stress in *PPM1D*-mutant cells. Altogether, our results demonstrate a role for SOD1 in the survival of *PPM1D*-mutant leukemia cells and highlight a new potential therapeutic strategy against *PPM1D*-mutant cancers.

## eLife assessment

Gain-of-function mutations and amplifications of PPM1D are found across several human cancers and are associated with advanced tumor stage and worse prognosis. Thus far, the clinical translation has not been possible due to the lack of PPM1D inhibitors with favorable pharmacokinetic properties. This **useful** study leverages CRISPR/Cas9 screening to determine that loss of SOD1 and is synthetic lethal with PPM1D mutation in leukemia. The mechanistic analyses are still **incomplete**.

## Introduction

Cellular DNA is frequently damaged by both endogenous and exogenous factors (*Hoeijmakers, 2009*). Unresolved DNA damage can lead to genomic instability, which is a hallmark of aging and cancer (*Hanahan and Weinberg, 2011*). Cells have evolved intricate mechanisms to detect and repair DNA lesions. The DNA damage response (DDR) is a complex network of signaling pathways that coordinate various cellular processes initiated by p53, such as DNA repair (*Ciccia and Elledge, 2010*), cell cycle checkpoint activation (*Harper et al., 1993*), and apoptosis (*Yonish-Rouach et al., 1991*). However, upon resolution of DNA damage, the cell must terminate the DDR to avoid prolonged cell cycle arrest and apoptosis. One critical mechanism for DDR termination is the expression of protein phosphatase $Mg^{2+}/Mn^{2+}$-dependent 1D (PPM1D) (*Fiscella et al., 1997*), which is induced by p53 and plays a key role in attenuating the response. PPM1D is a member of the PP2C family of serine/threonine protein phosphatases and has been shown to dephosphorylate a wide range of DDR signaling molecules including p53, p38 MAPK, CHK1, CHK2, and H2AX (*Bulavin et al., 2002*; *Cha et al., 2010*; *Lu et al., 2005*; *Oliva-Trastoy et al., 2007*; *Takekawa et al., 2000*). These dephosphorylation events generally lead to reduced activity of the targets, ultimately resulting in deactivation of the DDR.

Dysregulation of PPM1D has been associated with the development of diverse cancers, including breast, ovarian, esophagus, brain, and others (*Khadka et al., 2022*; *Li et al., 2002*; *Li et al., 2020b*; *Ruark et al., 2013*; *Zhang et al., 2014*). *PPM1D* is located on chromosome 17q and therefore frequently amplified in breast and ovarian cancers exhibiting 17q23 amplifications (*Li et al., 2002*; *Ruark et al., 2013*). These amplifications result in overexpression of the wild-type (WT) PPM1D protein and consequently leads to suppression of p53 and other PPM1D targets in the DDR (*Bulavin et al., 2002*; *Lambros et al., 2010*). In addition, PPM1D can also become dysregulated through mutations in its terminal exon. These mutations produce a truncated protein that is stabilized, evading proteasome-mediated degradation (*Tokheim et al., 2021*). The resulting mutant protein maintains its phosphatase activity and is found at high levels even in the absence of DNA damage. Excessive PPM1D activity leads to constitutive dephosphorylation and downregulation of PPM1D targets including multiple members of the DDR (*Hsu et al., 2018*). These gain-of-function *PPM1D* mutations are observed in diverse solid cancers including osteosarcoma (*He et al., 2021*), colorectal carcinoma (*Peng et al., 2014*; *Yin et al., 2013*), diffuse midline gliomas (*Wang et al., 2011*; *Zhang et al., 2014*), and others. Moreover, *PPM1D* mutations and overexpression are associated with advanced tumor stage, worse prognosis, and increased lymph node metastasis (*Fu et al., 2014*; *Jiao et al., 2014*; *Li et al., 2020a*; *Li et al., 2020b*; *Peng et al., 2014*; *Zhang et al., 2014*).

More recently, *PPM1D* mutations have been shown to drive expansion of hematopoietic stem cells (*Bolton et al., 2020*; *Hsu et al., 2018*; *Kahn et al., 2018*) in association with clonal hematopoiesis (CH), a pre-malignant state associated with an increased risk of hematologic malignancies and elevated all-cause mortality (*Genovese et al., 2014*; *Jaiswal et al., 2014*). *PPM1D* mutations are particularly enriched in patients with prior exposure to cytotoxic therapies, who have a high risk of therapy-related myeloid neoplasms (t-MN) (*Hsu et al., 2018*; *Lindsley et al., 2017*). Given the prevalence of *PPM1D* aberrations in cancer, PPM1D is an attractive therapeutic target. Ongoing efforts are focused on elucidating the structure of PPM1D to improve drug design and development (*Miller et al., 2022*). While several inhibitors thus far have shown efficacy in vitro, few have been studied in vivo and none have progressed to clinical trials due to poor bioavailability. Therefore, identifying targetable, synthetic-lethal partners to exploit the genetic defects of *PPM1D*-altered cells can offer an alternative therapeutic approach.

In this study, we performed an unbiased, whole-genome CRISPR screen to investigate genes essential for cell survival in *PPM1D*-mutated leukemia cell lines. We identified superoxide dismutase-1 (*SOD1*) as a novel synthetic-lethal dependency of PPM1D which was validated by genetic and

pharmacological approaches. We showed that the mutant cells display compromised responses to oxidative stress and DNA damage, leading to increased reactive oxygen species (ROS) and genomic instability. These results provide valuable insights into the biological processes corrupted by mutant PPM1D and underscore the potential of SOD1 as a targetable vulnerability in this context.

## Results

### *SOD1* is a synthetic-lethal vulnerability of *PPM1D*-mutant leukemia cells

CRISPR dropout screens have emerged as a powerful tool to assess the functional importance of individual genes within a particular pathway by measuring the impact of their depletion on cell viability or fitness. To identify genes essential for *PPM1D*-mutant cell survival, we first created isogenic WT and *PPM1D*-mutant Cas9-expressing OCI-AML2 leukemia cell lines and selected two *PPM1D*-mutant clones for CRISPR screening (*Figure 1—figure supplement 1A*). We transduced the cells with a whole-genome lentiviral library containing 90,709 guide RNAs (gRNAs) targeting 18,010 human genes (*Tzelepis et al., 2016*). At day 10 post-transduction, the cells were harvested for the first time-point and then subsequently passaged for an additional 18 days to allow for negatively selected gene-knockout cells to 'drop out'. The remaining pool of cells were collected for deep sequencing analysis of gRNA abundance (*Figure 1A*). We analyzed genes that were specifically depleted in the mutant but not WT cells using the MaGECK-VISPR pipeline (*Li et al., 2014*). Differentially depleted genes are those for which the knockout or depletion of the gene results in a significant impact on the viability or growth of *PPM1D*-mutant cells compared to WT control cells. Through this analysis, we identified 409 differentially depleted genes in one of the *PPM1D*-mutant clones and 92 differentially depleted genes in the other clone while adhering to the maximum false discovery rate (FDR) cutoff of 25%. Among these genes, we found 37 common candidates that were depleted in both *PPM1D*-mutant biological replicates that were not depleted in the WT control cells (*Figure 1—figure supplement 1B*, *Figure 1—source data 1*).

Gene ontology analysis of these top essential genes demonstrated an enrichment in pathways related to DNA repair, interstrand crosslink (ICL) repair, and cellular responses to stress (*Figure 1B*). Pathway analyses with the KEGG and REAC databases revealed a significant enrichment of the Fanconi anemia (FA) repair pathway, with notable genes such as *BRIP1* (*FANCJ*), *FANCI*, *FANCA*, *SLX4* (*FANCP*), *UBE2T* (*FANCT*), and *C19orf40* (*FAAP24*) (*Figure 1—figure supplement 1C*). Interestingly, our dropout screen revealed that superoxide dismutase (SOD) [Cu/Zn], or *SOD1*, was the top essential gene based on fitness score (*Figure 1C*). SOD1 is a crucial enzyme involved in scavenging super-oxide ($O_2^-$) radicals, which are harmful byproducts of mitochondrial cellular metabolism. Excessive ROS causes oxidative stress, which can damage cellular structures including DNA, proteins, and lipids. SOD1 is an attractive therapeutic target due to the availability of SOD1 small-molecule inhibitors that are being tested in clinical trials (*Lin et al., 2013*; *Lowndes et al., 2008*). Therefore, we decided to further investigate the role of SOD1 in promoting *PPM1D*-mutant cell survival.

To validate the essentiality of SOD1 in *PPM1D*-mutant cells, we performed in vitro competitive proliferation assays in two different acute myeloid leukemia (AML) cell lines, OCI-AML2 and OCI-AML3. We transduced isogenic WT and *PPM1D*-mutant Cas9-expressing cells with either empty vector (EV) or sg*SOD1*-expressing lentiviral vectors containing a blue fluorescent protein (BFP) reporter. We validated the loss of SOD1 protein expression by western blot (*Figure 1—figure supplement 1D*) and confirmed that transduction of the EV control did not alter cellular fitness (*Figure 1—figure supplement 1E*). While loss of *SOD1* had minimal effects on the fitness of WT cells, *PPM1D*-mutant cells with *SOD1* deletion had significant reduction in cellular growth in both OCI-AML2 and OCI-AML3 cells in vitro (*Figure 1D*).

To test if *SOD1* deletion affected the fitness of *PPM1D^mut^* vs WT leukemia cells in vivo, we transplanted *PPM1D*-mutant and -WT OCI-AML2 cells with or without *SOD1* deletion into immunodeficient (NSG) mice. Mice transplanted with control *PPM1D*-mutant and -WT cells (with intact SOD1) had a similar median survival of 32 days. When *SOD1* was deleted, the survival of mice transplanted with PPM1D-WT leukemia cells increased to a median of 43 days. Importantly, the survival of mice transplanted with *PPM1D^mut^*-*SOD1^−/−^* cells was even more significantly extended to a median time of 55 days (*Figure 1E*). These data provide an in vivo validation of the CRISPR screen demonstrating a differential dependency between *PPM1D*-mutant vs -WT cells on SOD1. Broadly, these results show

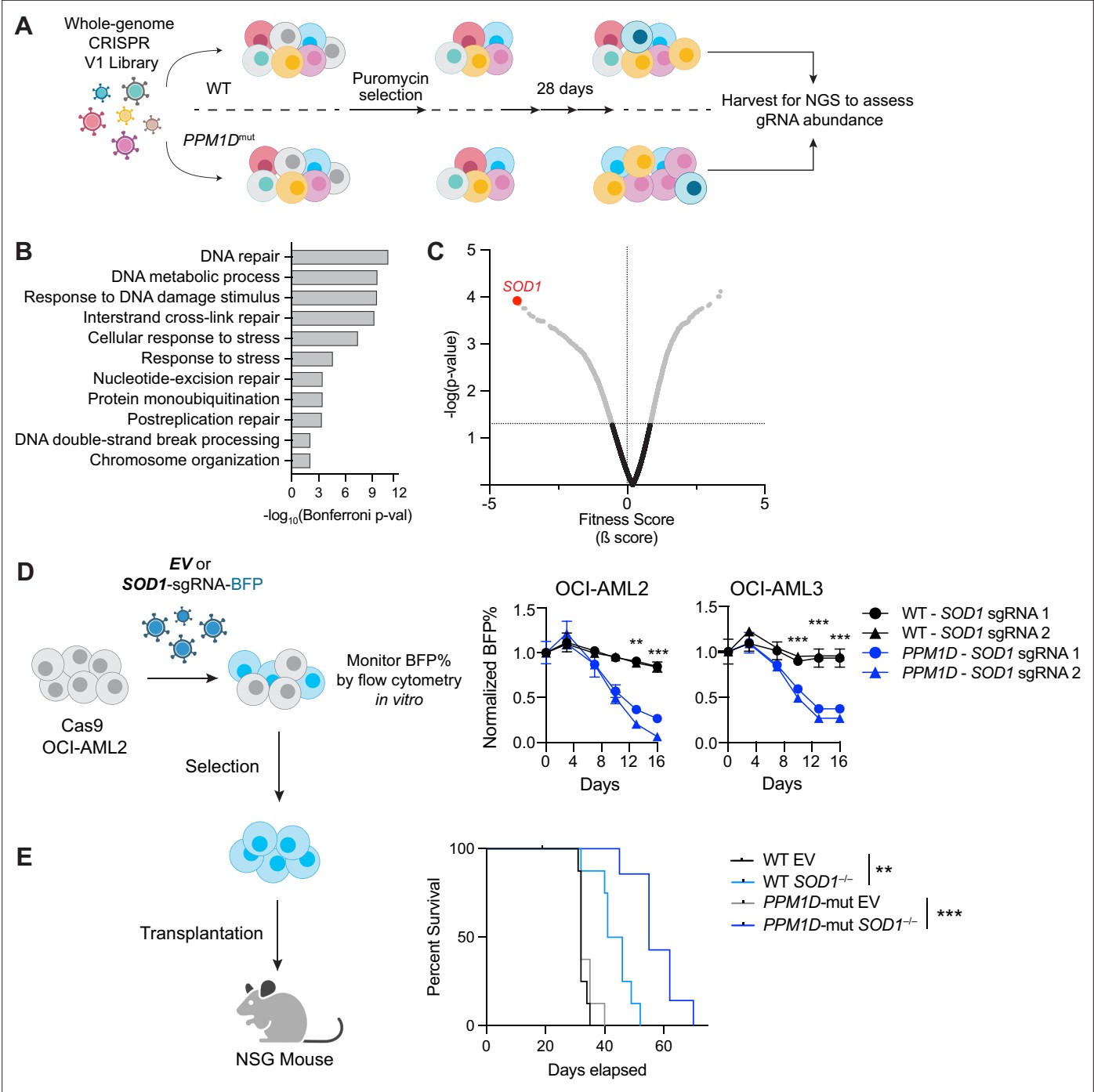

**Figure 1.** *SOD1* is a synthetic-lethal vulnerability of *PPM1D*-mutant leukemia cells. (**A**) Schematic of whole-genome CRISPR dropout screen. Wild-type (WT) Cas9-expressing OCI-AML2 and two isogenic *PPM1D*-mutant lines were transduced with the Human Improved Whole Genome Knockout CRISPR library V1 containing 90,709 guide RNAs (gRNAs) targeting 18,010 human genes at low multiplicity of infection (MOI~0.3). Each condition was performed in technical triplicates. Three days post-transduction, cells underwent puromycin selection for 3 days. Cells were harvested at day 10 as the initial timepoint and then harvested every 3 days afterward. sgRNA-sequencing was performed on cells collected on day 28. (**B**) Top biological processes based on gene ontology analysis of the top 37 genes essential for *PPM1D*-mutant cell survival. Enrichment and depletion of guides and genes were analyzed using MAGeCK-VISPR by comparing read counts from each *PPM1D*-mutant cell line replicate with counts from the initial starting population at day 10. (**C**) Volcano plot of synthetic-lethal hits ranked by fitness score with a negative score indicating genes for which their knockout leads to decreased growth or survival. *SOD1* (highlighted) was the top hit from the screen. (**D**) Left: Schematic of competitive proliferation assays used for validation of CRISPR targets. Right: WT and *PPM1D*-mutant Cas9-OCI-AML2 and Cas9-OCI-AML3 cells were transduced with lentiviruses containing a single *SOD1*-gRNA with a blue fluorescent protein (BFP) reporter. Cells were assayed by flow cytometry every 3–4 days and normalized

*Figure 1 continued on next page*

*Figure 1 continued*

to the BFP percentage at day 3 post-transduction. Two unique gRNAs against *SOD1* were used per cell line and each condition was performed in technical duplicates; multiple unpaired t-tests, **p<0.01, ***p<0.001. (**E**) Left: Cas9-expressing WT and *PPM1D*-mutant cells were transduced with control or sg*SOD1*-containing lentiviruses and underwent puromycin (3 μg/mL) selection for 3 days prior to transplantation. Sublethally irradiated (250 cGy) NSG mice were intravenously transplanted with $3 \times 10^6$ cells. Right: Kaplan-Meier survival curve of mice transplanted with WT or *PPM1D*-mutant (gray) leukemia cells with or without *SOD1* deletion. The median survival of mice transplanted with WT, WT/*SOD1*$^{-/-}$, *PPM1D*$^{mut}$, and *PPM1D*$^{mut}$/*SOD1*$^{-/-}$ leukemia cells was 32, 43, 32, and 55 days, respectively; Mantel-Cox test, **p<0.01, ***p<0.001.

The online version of this article includes the following source data and figure supplement(s) for figure 1:

**Source data 1.** CRISPR dropout screen raw data and top 37 gene candidates.

**Figure supplement 1.** *SOD1* is a synthetic-lethal vulnerability of *PPM1D*-mutant leukemia cells.

**Figure supplement 1—source data 1.** Western blot validation of OCI-AML2 PPM1D-mutant clones after CRISPR editing.

**Figure supplement 1—source data 2.** Western blot validation of SOD1 deletion in WT and PPM1D-mutant cells.

that loss of *SOD1* confers a disadvantage to leukemia cells that is markedly amplified in the context of the PPM1D-truncating mutation.

## *PPM1D*-mutant cells are sensitive to SOD1 inhibition and have increased oxidative stress

We next wanted to test if pharmacological inhibition of SOD1 could mimic the genetic deletion of *SOD1*. We used two different SOD1 inhibitors, 4,5-dichloro-2-*m*-tolyl pyridazin-3(2*H*)-one (also known as lung cancer screen-1 [LCS-1]) and *Bis*-choline tetrathiomolybdate (ATN-224), which work by different mechanisms. LCS-1 is a small molecule that binds to SOD1 and disrupts its activity (**Somwar et al., 2011**), while ATN-224 is a copper chelator that reduces SOD1 activity by decreasing the availability of copper ions, which are an essential SOD1 cofactor (**Juarez et al., 2006**).

To study the sensitivity of the mutant cells to SOD1 inhibition, we engineered truncating *PPM1D* mutations into three patient-derived AML cell lines, MOLM-13, OCI-AML2, and OCI-AML3, which harbor distinct genetic backgrounds and AML driver mutations. At baseline, we found that *PPM1D*-mutant cells had increased SOD activity compared to WT cells and confirmed that SOD activity was significantly inhibited upon treatment with ATN-224 in a dose-dependent manner (**Figure 2—figure supplement 1A**). In addition, ATN-224 induced a significantly greater proportion of apoptotic *PPM1D*-mutant than *PPM1D*-WT cells (**Figure 2—figure supplement 1B**). *PPM1D*-truncating mutations conferred significant sensitivity to SOD1 inhibition compared to their WT counterparts in all three AML cell lines (**Figure 2A**, **Figure 2—figure supplement 2A**). To determine if this cytotoxicity was dependent on oxidative stress, we treated the cells with SOD1 inhibitors in combination with an antioxidant, *N*-acetylcysteine (NAC). Importantly, NAC supplementation was able to completely rescue the sensitivity of mutant cells to both LCS-1 and ATN-224 treatment (**Figure 2B**, **Figure 2—figure supplement 1C**), suggesting that ROS generation contributes to the sensitivity of mutant cells to SOD1 inhibition.

Activating mutations in oncogenes often lead to increased ROS generation by altering cellular metabolism, inducing replication stress, or dysregulating redox homeostasis (**Maya-Mendoza et al., 2015**; **Park et al., 2014**). We therefore hypothesized that *PPM1D*-mutant cells have increased oxidative stress, leading to reliance on SOD1 for protection. SOD1 catalyzes the breakdown of superoxide into hydrogen peroxide and water. Therefore, we assessed cytoplasmic and mitochondrial superoxide levels using dihydroethidium and MitoSOX Green, respectively. These fluorogenic dyes are rapidly oxidized by superoxide, but not other types of ROS, to produce green fluorescence. We observed that in the absence of exogenous stressors, *PPM1D*-mutant cells had a moderate increase in superoxide radicals (**Figure 2C**, **Figure 2—figure supplement 1C**). SOD2 is the primary superoxide dismutase in the mitochondria responsible for catalyzing superoxide into $H_2O_2$. Given the increase in mitochondrial superoxide levels, we assessed levels of SOD2 protein levels. Surprisingly, there were no baseline differences or compensatory changes in SOD2 after *SOD1* deletion (**Figure 2—figure supplement 2C**).

Free radicals can be detrimental to cells due to their ability to oxidize proteins, lipids, and DNA. Therefore, we also measured levels of lipid peroxidation as an additional measure of oxidative stress. Consistent with the increase in superoxide radicals, we observed a concurrent increase in

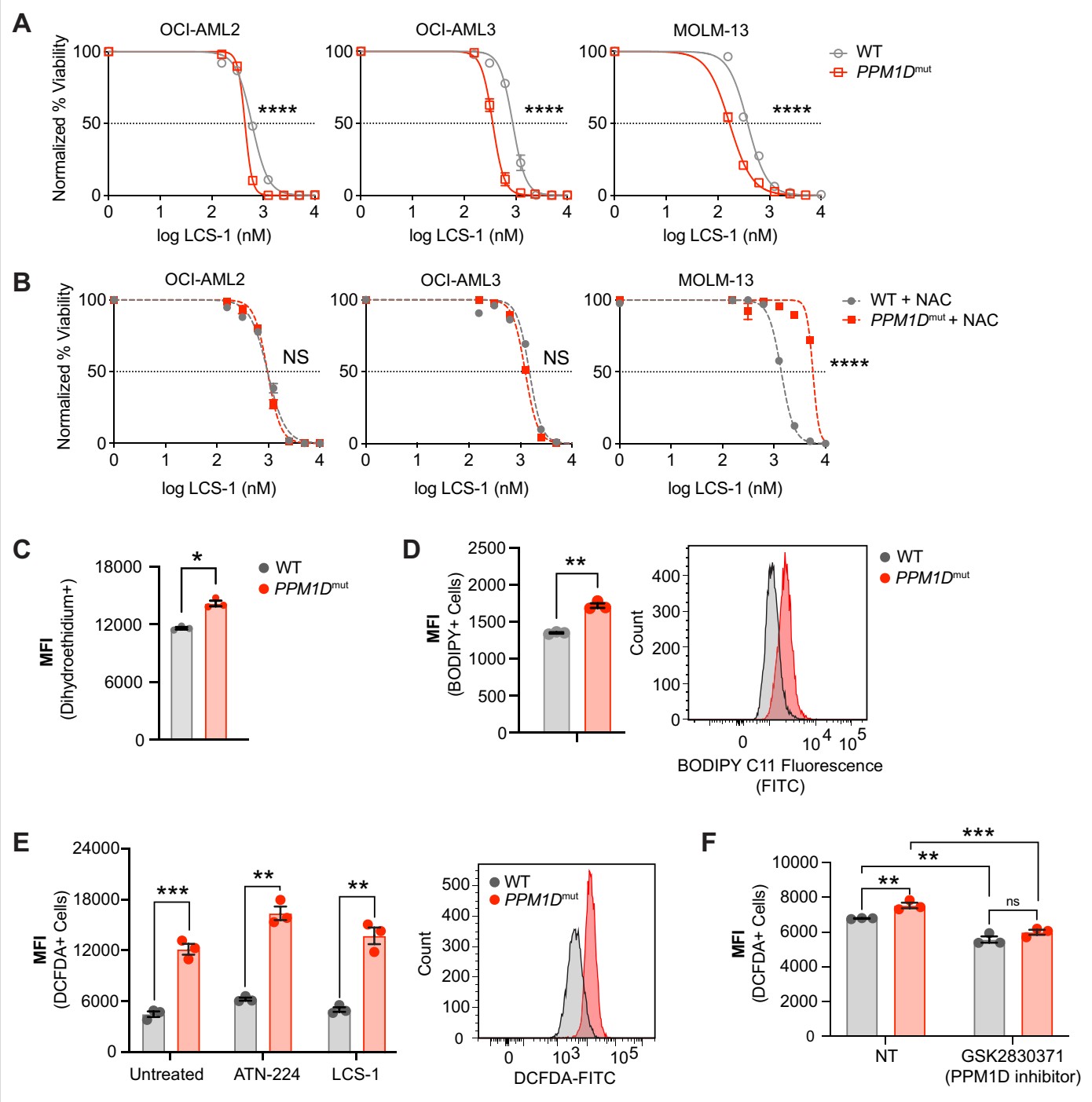

**Figure 2.** *PPM1D*-mutant cells are sensitive to SOD1 inhibition and have increased oxidative stress. (**A,B**) Dose response curves for cell viability with SOD1-inhibitor (LCS-1) (**A**) or LCS-1 in combination with 0.25 uM NAC (**B**) in WT and *PPM1D*-mutant leukemia cell lines after 24-hours. Mean ± SD (n=3) is shown with a non-linear regression curve. All values are normalized to the baseline cell viability with vehicle, as measured by MTT assay. (**C**) Endogenous cytoplasmic superoxide levels of WT and *PPM1D*-mutant leukemia cell lines were measured using dihydroethidium (5 uM). The mean fluorescence intensity (MFI) of dihydroethidium was measured by flow cytometry. Mean ± SD (n=3) is shown. (**D**) Lipid peroxidation measured using BODIPY 581/591 staining (2.5 uM) of WT and *PPM1D*-mutant OCI-AML2 cells. The MFI was measured by flow cytometry. Mean ± SD (n=3) is shown. (**E-F**) Measure of total reactive oxygen species using 2',7'–dichlorofluorescin diacetate (DCFDA) staining (10 uM) measured by flow cytometry. WT and *PPM1D*-mutant OCI-AML2 cells were measured at baseline and 24-hrs after SOD1 inhibition (ATN-224 12.5 uM, LCS-1 0.625 uM) (**E**) or 24-hrs after pharmacologic PPM1D inhibition (GSK2830371, 5 uM) (**F**); unpaired t-tests were used for statistical analyses, ns=non-significant (p>0.05), **p<0.01, ***p<0.001, ****p<0.0001.

*Figure 2 continued on next page*

*Figure 2 continued*

The online version of this article includes the following source data and figure supplement(s) for figure 2:

**Figure supplement 1.** *PPM1D*-mutant cells have increased oxidative stress.

**Figure supplement 2.** *PPM1D*-mutant cells have increased oxidative stress.

**Figure supplement 2—source data 1.** Western blot of SOD2 expression at baseline and after SOD1 deletion.

lipid peroxidation in the *PPM1D*-mutant cells (**Figure 2D**). Using 2′7′-dichlorofluorescein diacetate (DCFDA) staining to measure total ROS levels, we observed that *PPM1D*-mutant cells harbored more total ROS compared to WT cells (**Figure 2E**).

To investigate whether the observed elevated ROS was a characteristic of other *PPM1D*-mutant cell lines, we measured ROS levels in two different germline models. Humans with germline mutations in *PPM1D* were first described by Jansen et al. in 2017 in patients with intellectual disability. This neurodevelopmental condition is named Jansen-de Vries syndrome (JdVS, OMIM #617450) and is characterized by frameshift or nonsense mutations in the last or second-to-last exons of the *PPM1D* gene. These mutations result in functionally active, truncated mutant proteins like those exhibited in human cancers and CH. Lymphoblastoid cell lines (LCLs) were generated from these JdVS patients by *Jansen et al., 2017*; *Wojcik et al., 2023*.

In addition to human *PPM1D*-mutant LCLs, we also generated mouse embryonic fibroblasts (MEFs) from a germline mouse model harboring a heterozygous truncating mutation in the terminal exon of *Ppm1d* (*Hsu et al., 2018*). When we measured total ROS from both the JdVS LCLs and the *Ppm1d*-mutant MEFs compared to their WT counterparts, both mutant models exhibited greater levels of total ROS (**Figure 2—figure supplement 2D and E**). Additionally, *PPM1D*-mutant LCLs were also more sensitive to pharmacological SOD1 inhibition compared to the WT LCL, GM12878 (**Figure 2— figure supplement 2F**). These results demonstrate that *PPM1D* mutations not only increase ROS in the context of cancer, where cellular metabolism is often altered, but can also alter redox homeostasis in non-transformed cells. Lastly, to determine if mutant PPM1D was associated with ROS generation, we treated isogenic OCI-AML2 WT and *PPM1D*-mutant cells with a PPM1D inhibitor, GSK2830371, for 24 hr. We found that pharmacological inhibition of PPM1D mildly decreased ROS levels in both WT and *PPM1D*-mutant cells (**Figure 2F**). Altogether, these data suggest a link between PPM1D and ROS production that results in mutant-specific cytotoxicity to SOD1 inhibition.

### *PPM1D*-mutant leukemia cells have altered mitochondrial function

Mitochondria are the primary source of ROS within the cell, as the electron transport chain is a major site of ROS production during oxidative phosphorylation. We next asked whether the observed increase in ROS in *PPM1D*-mutant cells was due to differences in mitochondrial abundance. We used two independent methods to measure mitochondrial mass, including MitoTracker Green flow cytometry (**Figure 3A**) and western blot analysis of mitochondrial complex proteins (**Figure 3B**). However, we did not observe a difference in mitochondrial mass with either method. This finding suggests that mechanisms other than a change in mitochondrial abundance are responsible for the increase in ROS levels in mutant cells, such as alterations in mitochondrial metabolism or changes in ROS scavenging systems.

To assess mitochondrial function, we performed seahorse assays in WT and *PPM1D*-mutant cells. Our seahorse assays revealed that the mutant cells have decreased mitochondrial respiration, as indicated by decreased basal, maximal, and ATP-linked respiration (**Figure 3C**). While *PPM1D*-mutant MOLM-13 and OCI-AML3 cells also had decreased basal respiration, there were variable differences in maximal and ATP-linked respiration compared to WT, suggesting possible cell line differences affecting mitochondrial respiration (**Figure 3—figure supplement 1A and B**). In addition to analyzing respiratory capacity, we also examined mitochondrial membrane potential (MMP) using the fluorescent dye MitoTracker CMXRos, which accumulates in the mitochondria in an MMP-dependent manner. We stained both WT and mutant cells with MitoTracker CMXRos and observed a decrease in MMP in the mutant cells (**Figure 3D**). Tracking cell numbers between the WT and mutant cell lines over time established this decrease in MMP was not due to altered cellular growth rates (**Figure 3—figure supplement 1C**). These findings, along with decreased respiratory capacity and increased mitochondrial ROS, indicate a mitochondrial defect in *PPM1D*-mutant cells.

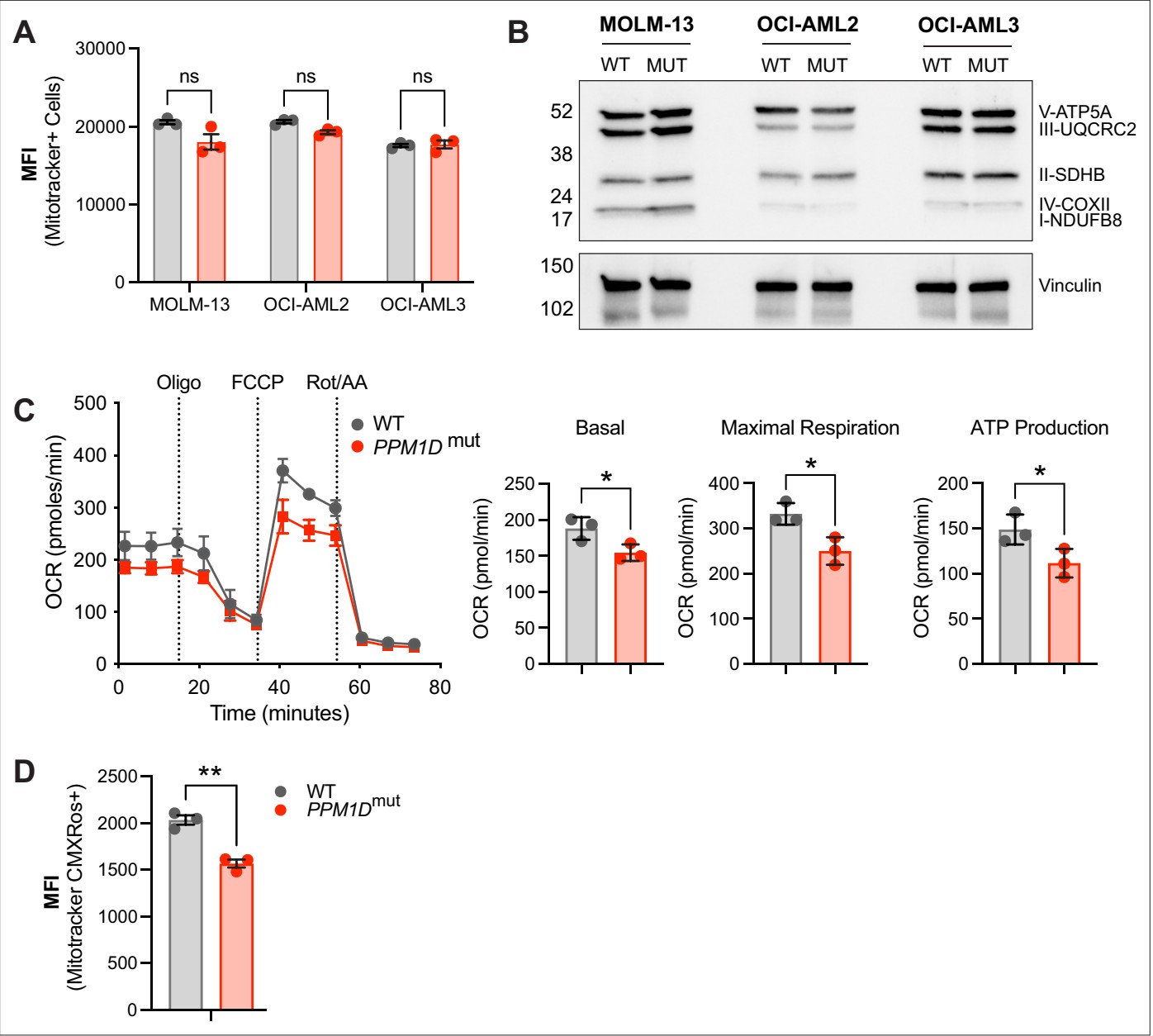

**Figure 3.** *PPM1D*-mutant cells have altered mitochondrial function. (**A**) Mitochondrial mass of wild-type (WT) and *PPM1D*-mutant leukemia cells was determined using MitoTracker Green (100 nM) and the mean fluorescence intensity was analyzed by flow cytometry. Data represents mean ± SD of triplicates. At least three independent experiments were conducted with similar findings; unpaired t-tests. (**B**) Immunoblot of WT and *PPM1D*-mutant cell lysates probed with the human OXPHOS antibody cocktail (1:1000) and vinculin (1:2000). (**C**) Measurement of mitochondrial oxygen consumption rate (OCR) by seahorse assay in WT and *PPM1D*-mutant OCI-AML2 cells after treatment with oligomycin (1.5 μM), FCCP (0.5 μM), and rot/AA (0.5 μM). Quantification of basal, maximal, and ATP-linked respiration are shown. Data shown are the mean ± SD of technical triplicates. (**D**) Mitochondrial membrane potential of WT and *PPM1D*-mutant OCI-AML2 cells was measured using MitoTracker CMXRos (400 nM). The mean fluorescence intensity (MFI) was measured and analyzed by flow cytometry. Data represents mean ± SD of triplicates, unpaired t-test, ns = non-significant (p>0.05), *p<0.05, **p<0.01.

The online version of this article includes the following source data and figure supplement(s) for figure 3:

**Source data 1.** Western blot of mitochondrial proteins in WT and PPM1D-mutant cells.

**Figure supplement 1.** *PPM1D*-mutant cells have altered mitochondrial function.

## *PPM1D*-mutant cells have a reduced oxidative stress response

Mitochondrial dysfunction and increased ROS production are closely intertwined. On one hand, mitochondrial dysfunction leads to increased ROS production as a result of impaired oxidative phosphorylation and increased electron leakage (*Turrens, 2003*). On the other hand, sustained oxidative stress can directly damage mitochondrial components and mtDNA and compromise their function (*Wallace, 2005*). To better understand the molecular basis for the observed mitochondrial dysfunction and dependency on SOD1, we performed bulk RNA-sequencing (RNA-seq) on Cas9-expressing WT and *PPM1D*-mutant OCI-AML2 cells transduced with *SOD1*-sgRNA to induce *SOD1* deletion or the EV control (*Figure 4—figure supplement 1A*). Both EV and SOD1-sgRNA vectors were tagged with a BFP reporter to identify transduced cells. The cells were collected 10 days post-transduction, the timepoint at which we observed 50% reduction of the *SOD1* deletion cells during the in vitro proliferation assays, reasoning this would capture the effects of *SOD1* deletion on cellular and metabolic processes while avoiding excessive cell death.

Analysis of the RNA-seq data revealed 2239 differentially expressed genes, with 1338 downregulated genes and 901 upregulated genes in the mutant cells compared to WT cells at baseline (*Figure 4—source data 1*). Gene set enrichment analysis (GSEA) of the differentially expressed genes showed an upregulation in genes related to cell cycle (GO: 0007049), cell division (GO: 0051301), DNA replication (GO: 005513), and mitophagy (GO: 0000423) in the *PPM1D*-mutant cells (*Figure 4A*). Interestingly, there was a significant downregulation of pathways related to the regulation of the oxidative stress response (GO: 1902882, *Figure 4—figure supplement 1B*), ROS metabolic processes (GO: 0072593), and oxidation reduction (GO: 0055114). Following *SOD1* deletion, the WT cells displayed notable upregulation of pathways associated with cell cycling, chromosome organization, cell division, and DNA repair. In contrast, the mutant cells showed significant downregulation of these same pathways (*Figure 4—figure supplement 1C*). Intriguingly, upon *SOD1* deletion, the mutant cells exhibited an upregulation in response to oxidative stress (GO:0006979, *Figure 4—figure supplement 1D*). This finding suggests a reactive transcriptional response to the heightened ROS levels resulting from the loss of SOD1.

As PPM1D is a phosphatase that can directly modulate the activation state of proteins, we examined whether there were alterations in protein and phosphoprotein levels in *PPM1D*-mutant cells using reverse-phase protein array (RPPA) analysis, mirroring the experimental design used for bulk RNA-seq (*Figure 4—figure supplement 1A*). By focusing on differential protein expression between WT and *PPM1D*-mutant cells, we aimed to capture the post-translational regulatory events that could contribute to the mitochondrial dysfunction observed in the mutants. The RPPA analysis of over 200 (phospho-)proteins covering major signaling pathways identified 128 differentially expressed proteins between *PPM1D*-mutant and control WT OCI-AML2 cells (a panel of 264 proteins), with 67 downregulated proteins and 61 upregulated proteins (*Figure 4—figure supplement 2A*, *Figure 4—source data 2*). Notably, over-representation analysis showed that among the differentially expressed proteins, there was a significant enrichment in the 'Response to Oxidative Stress' pathway in the mutant cells (–log10(p-value)=24.164) compared to WT, with a particular emphasis on the downregulated proteins of this pathway (–log10(p-value)=15.457, *Figure 4B*, *Figure 4—source data 3*). While the RNA-seq suggested a transcriptional upregulation of the response to oxidative stress in the mutant cells after *SOD1* deletion, the RPPA data revealed that the mutant cells continued to exhibit decreased expression in proteins associated with the oxidative stress response (*Figure 4—figure supplement 2B*). Taken together, these findings suggest that *PPM1D*-mutant cells have an inherent impairment in their baseline response to oxidative stress.

To further explore the diminished oxidative stress response in the mutant cells, we assessed their total- and small-molecule-antioxidant capacity. Total antioxidant capacity refers to the overall ability of the cells to counteract free radicals and reduce oxidative damage. This includes enzymatic antioxidants such as catalase, SODs, and peroxidases. Small-molecule antioxidant capacity measures the capacity of low molecular weight antioxidants, such as glutathione (GSH) and vitamin E, to neutralize ROS (*Hawash et al., 2022*). Our results showed that *PPM1D*-mutant cells have significantly reduced total- and small-molecule antioxidant capacity compared to WT cells (*Figure 4C*).

Subsequently, we measured intracellular GSH, a pivotal antioxidant crucial for maintaining cellular redox balance and protecting against oxidative stress. Strikingly, our analysis revealed a higher proportion of mutant cells with diminished GSH levels compared to their WT counterparts (*Figure 4D*). We

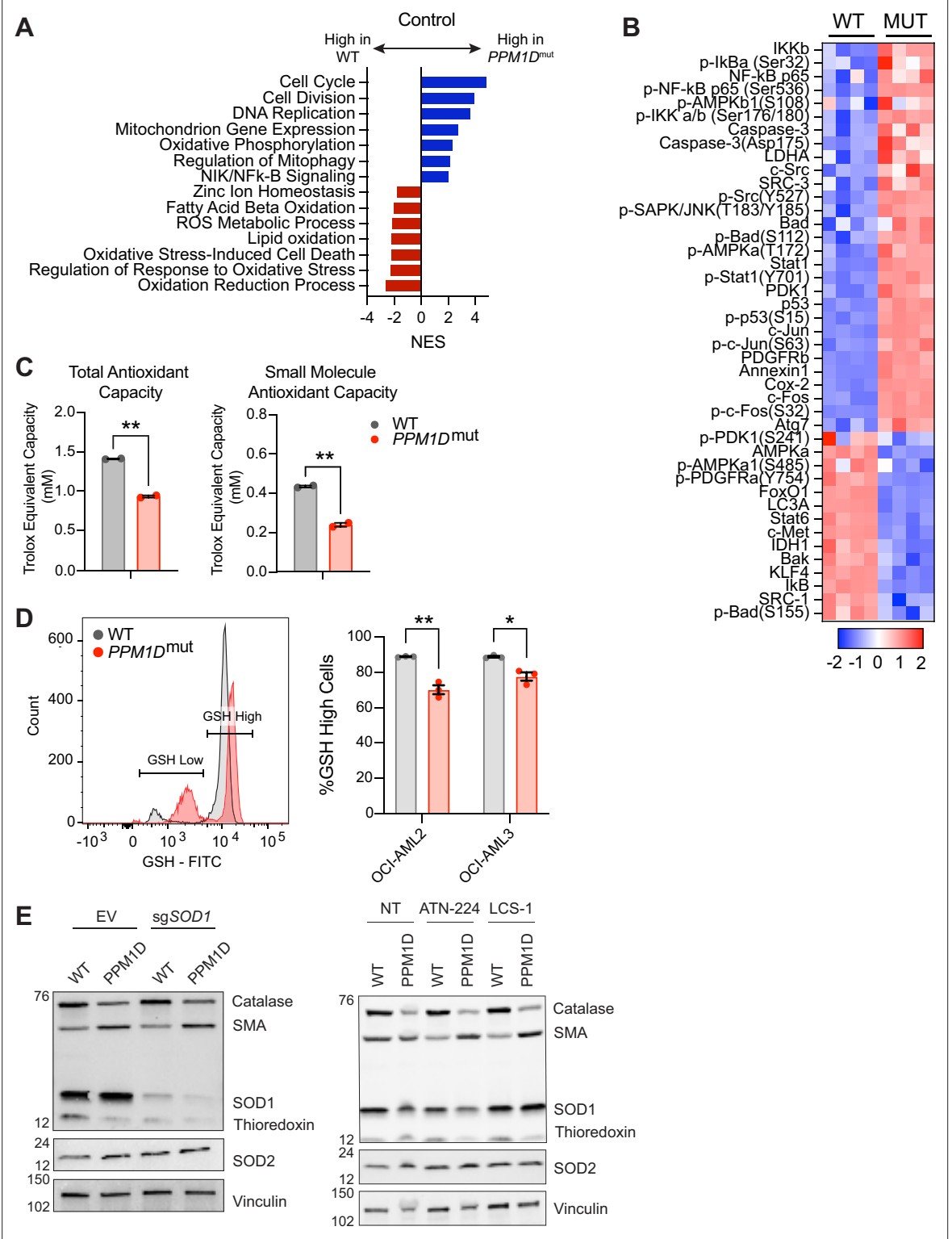

**Figure 4.** *PPM1D*-mutant cells have a reduced oxidative stress response. (**A**) RNA-sequencing (RNA-seq) gene set enrichment analysis (GSEA) of *PPM1D*-mutant cells compared to wild-type (WT) Cas9-OCI-AML2 cells. Significantly up- and downregulated pathways are indicated by the blue and red bars, respectively. Normalized enrichment scores (NES) are shown with false discovery rate (FDR) < 0.25. (**B**) Reverse-phase protein array (RPPA) profiling of WT and *PPM1D*-mutant OCI-AML2 cells. Proteins from the 'Response to Oxidative Stress' pathway have been selected for the heatmap. Each column represents a technical replicate. See *Figure 4—source data 2* for the raw data. (**C**) Total- and small-molecule antioxidant capacity of WT and *PPM1D*-mutant cells performed in technical duplicates. (**D**) Intracellular glutathione (GSH) levels measured by flow cytometry using the Intracellular

*Figure 4 continued on next page*

*Figure 4 continued*

GSH Detection Assay Kit (Abcam). Left: Representative flow cytometry plot demonstrating the gating for GSH-high and GSH-low populations. Right: Quantification of the percentage of GSH-high cells for each cell line. Mean ± SEM (n=3) are shown. (**E**) Immunoblot of WT and *PPM1D*-mutant OCI-AML2 after transduction with the empty vector (EV) control and after *SOD1* deletion (left) or after treatment with SOD1 inhibitors for 16 hr (right, ATN-224 12.5 µM, lung cancer screen-1 [LCS-1] 1.25 µM). Lysates were probed with an anti-oxidative stress defense cocktail (1:250), SOD2 (1:1000), and vinculin (1:2000). SMA = smooth muscle actin. Student's t-tests were used for statistical analysis; **p<0.01, *p<0.05.

The online version of this article includes the following source data and figure supplement(s) for figure 4:

**Source data 1.** RNA-seq gene expression analysis of WT and PPM1D-mutant cells after transduction with empty vector [EV] or sgSOD1 lentiviruses.

**Source data 2.** Reverse phase protein array (RPPA) analysis of WT and PPM1D-mutant cells at baseline and after SOD1-deletion.

**Source data 3.** Reverse phase protein array (RPPA) over-representation analysis pathways.

**Source data 4.** Western blot analysis of oxidative stress defense proteins after genetic deletion and pharmacologic inhibition of SOD1.

**Figure supplement 1.** *PPM1D*-mutant cells have reduced oxidative stress response.

**Figure supplement 2.** *PPM1D*-mutant cells have reduced oxidative stress response.

also measured the protein levels of key antioxidant enzymes by western blot. While we saw similar protein levels of SOD1 in both WT and mutant cells, we observed a reduction in the thioredoxin and catalase levels (*Figure 4E*). These results provide evidence to support the RNA-seq and RPPA findings that *PPM1D*-mutant cells have impaired antioxidant defense mechanisms, leading to an elevation in ROS levels.

## *PPM1D* mutations increase genomic instability and impair non-homologous end-joining repair

In addition to a decreased response to oxidative stress, the RNA-seq GSEA also revealed differential responses to DNA repair. Upon *SOD1* deletion, WT cells significantly upregulated the regulation of DNA repair (GO:0006281), double-stranded break (DSB) repair (GO:0006302), homologous recombination (HR) (GO:0035825), and more. However, there was a striking downregulation of DNA repair pathways after deletion of *SOD1* in the mutant cells (*Figure 4—figure supplement 1C*). PPM1D plays a key role in suppressing the DDR by dephosphorylating, thereby inactivating, p53 and other key upstream and downstream effectors of the pathway. Truncating mutations and amplifications in *PPM1D* that lead to increased PPM1D activity may therefore inhibit DNA damage repair and increase genomic instability. Oxidative stress and ROS also pose endogenous challenges to genomic integrity. Therefore, we hypothesized that due to the increase in ROS within the mutant cells, loss of *SOD1* may lead to unsustainable accumulation of DNA damage and overwhelm the mutant cell's DNA repair capacity.

To test this hypothesis, we first sought to establish the baseline levels of DNA damage in *PPM1D*-altered cells. We performed alkaline comet assays in MEFs and found a significant increase in single- and double-stranded DNA breaks in mutant cells compared to WT (*Figure 5A*). As ROS are known to contribute to oxidative DNA damage, we further assessed the levels of 8-oxo-2'-deoxyguanosine, a well-established marker of oxidative DNA damage. Strikingly, the mutant cells demonstrated elevated levels of oxidative DNA damage at baseline (*Figure 5B*). We also performed metaphase spreads in mouse primary B-cells to investigate chromosomal aberrations, which are consequences of abnormal DSB repair. WT and *Ppm1d*-mutant mouse primary resting CD43+ B-cells were purified from spleens and stimulated with LPS, IL-4, and CD180 to induce proliferation. The cells were then treated with either low- or high-dose cisplatin for 16 hr. Consistent with our comet assay findings, we observed that *Ppm1d*-mutant cells harbored approximately twofold more chromosomal breaks per metaphase after exposure to cisplatin (*Figure 5C*). When we classified the chromosomal aberrations into subtypes, we observed that the mutant cells had increased numbers of each type of aberration. These results demonstrate that mutations in *PPM1D* increase genomic instability.

To further assess the DNA repair efficiency of *PPM1D*-mutant cells, we utilized U2OS DNA repair reporter cell lines which express a green fluorescent protein (GFP) cassette when specific DNA repair pathways are active after stimulation when the I-SceI restriction enzyme is induced to stimulate a DSB. To test for HR, tandem defective GFP genes can undergo HR to generate GFP+ cells. Non-homologous end-joining (NHEJ) repairs a defective GFP in a distinct cassette (*Weinstock et al., 2006*). Because the U2OS parental line harbors an endogenous heterozygous *PPM1D*-truncating mutation

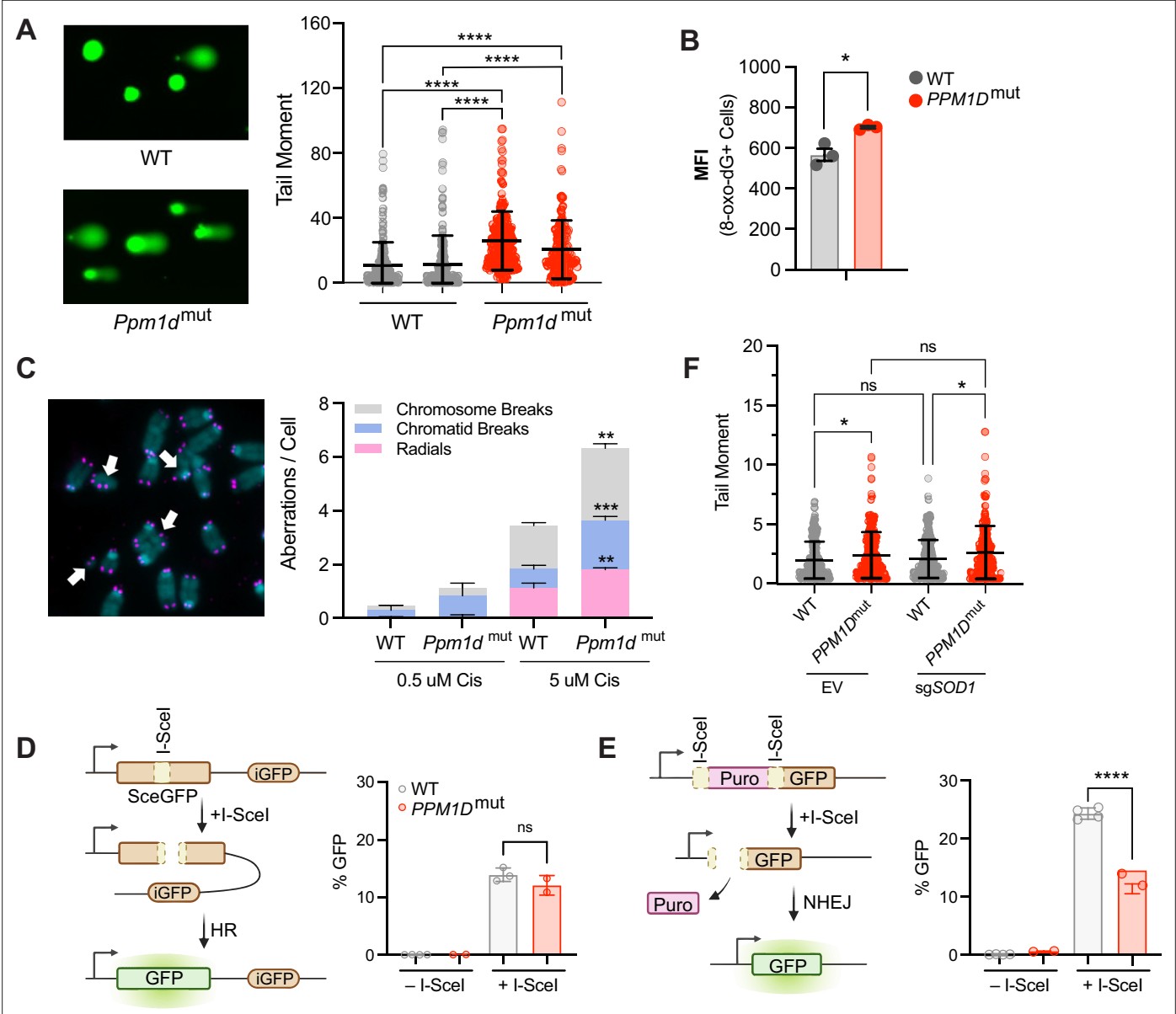

**Figure 5.** *PPM1D* mutations increase genomic instability and impair non-homologous end-joining. (**A**) Left: Representative images of comet assays of mouse embryonic fibroblasts (MEFs). Two biological replicates were assessed for each genotype. Right: Quantification of n≥150 comets per experimental group with the Comet IV software; two-way ANOVA. (**B**) Mean fluorescent intensity (MFI) of 8-oxo-2'-deoxyguanosine (8-oxo-dG) lesions within wild-type (WT) and *PPM1D*-mutant OCI-AML2 cells as measured by flow cytometry; Student's t-test. (**C**) Left: Representative images of metaphase spreads of WT and *Ppm1d*-mutant mouse primary B-cells treated with low (0.5 µM) or high (5 µM) doses of cisplatin. Right: n≥50 metaphase cells were quantified in each experimental condition for chromosomal aberrations (white arrows). n=2 biological replicates used for each genotype. Student's t-test was used for statistical analysis. (**D–E**) Left: Schematic of the homologous recombination (**D**) or non-homologous end-joining (**E**) U2OS DNA damage repair cassettes. Right: Quantification of GFP% analyzed by flow cytometry 48 hr after induction of DNA damage by I-SceI transduction; Student's t-test. (**F**) Comet assay quantification of WT and *PPM1D*-mutant Cas9-OCI-AML2 cells 6 days after lentiviral transduction with the empty vector (EV) control, or sg*SOD1* to induce *SOD1* deletion. Quantification and analyses of tail moments were performed using the Comet IV software. n≥150 comets were scored per experimental group; two-way ANOVA. Data are mean ± SD (n=3), ns = non-significant (p>0.05), *p<0.05, **p<0.01, ***p<0.001, ****p<0.0001.

The online version of this article includes the following source data and figure supplement(s) for figure 5:

**Source data 1.** Comet assay assessing baseline levels of DNA damage in WT and Ppm1d-mutant mouse embryonic fibroblasts.

**Source data 2.** Metaphase spread of WT and Ppm1d-mutant mouse primary B-cells after treatment with cisplatin.

**Figure supplement 1.** *PPM1D*-mutations increase genomic instability and impairs non-homologous end-joining repair.

**Figure supplement 1—source data 1.** Western blot analysis of CRISPR-edited U2OS clones validating the correction of the endogenous PPM1D

*Figure 5 continued on next page*

*Figure 5 continued*

mutations to the wild type form.

**Figure supplement 1—source data 2.** Immunofluorescence microscopy of WT and Ppm1d-mutant mouse embryonic fibroblasts stained with Rad51.

**Figure supplement 1—source data 3.** Immunofluorescence microscopy of WT and Ppm1d-mutant mouse embryonic fibroblasts stained with 53BP1.

(R458X) (*Kleiblova et al., 2013*), we corrected the lines to generate the isogenic *PPM1D* WT control (*Figure 5—figure supplement 1A*).

With two isogenic clones for each reporter cell line, we transfected the *PPM1D*-WT and -mutant U2OS clones with I-SceI and measured GFP expression by flow cytometry after 48 hr. Our results showed similar levels of HR-mediated repair in both WT and mutant clones (*Figure 5D*). Prior studies have shown that WT PPM1D promotes HR by forming a stable complex with BRCA1-BARD1, thereby enhancing their recruitment to DSB sites (*Burdova et al., 2019*). Although gain-of-function mutations in PPM1D lead to persistent PPM1D activity, it may not necessarily result in increased HR repair. Several factors can limit the extent of HR enhancement. For instance, HR is typically restricted to the S/G2 phase of the cell cycle and is a multi-step process that beings with DNA end resection (*Xu and Xu, 2020*). This is a crucial initial step that generates single-stranded DNA overhangs to facilitate strand invasion and recombination (*Gnügge and Symington, 2021*). Therefore, the impact of mutant PPM1D on HR may be constrained by the efficiency of DNA end resection and cell cycling, among other regulatory mechanisms within the HR pathway.

In contrast, we saw significantly decreased NHEJ repair in the *PPM1D*-mutant clones (*Figure 5E*). This downregulation of NHEJ may be due to diminished activation of yH2AX and ataxia telangiectasia mutated (ATM). These two proteins serve as key upstream regulators within the DDR and are subject to dephosphorylation by PPM1D (*Cha et al., 2010*; *Lu et al., 2005*). In addition, prior studies have also shown that PPM1D modulates lysine-specific demethylase 1 activity, which is important for facilitating the recruitment of 53BP1 to DNA damage sites through RNF168-dependent ubiquitination (*Peng et al., 2015*). *PPM1D* mutations may therefore lead to impairment of NHEJ through dysregulation of 53BP1 recruitment. To confirm this, we performed immunofluorescence imaging of Rad51 and 53BP1 foci. The recruitment of Rad51 and 53BP1 to the sites of DNA damage are important for the activation of HR and NHEJ, respectively. We analyzed MEFs at baseline and after irradiation (10 Gy) and observed similar numbers of Rad51 foci in *Ppm1d*-mutant and WT cells (*Figure 5—figure supplement 1B*). In contrast, *Ppm1d*-mutant MEFs had fewer 53BP1 foci, indicating decreased NHEJ repair capacity that was consistent with our U2OS reporter line findings (*Figure 5—figure supplement 1C*). Comet assays were performed in parallel with the immunofluorescence experiments to show that the mutant cells had increased DNA damage (*Figure 5—figure supplement 1D*). Therefore, the decrease in foci was not due to resolution of DNA damage, but rather due to inefficient DNA repair.

In light of the elevated levels of DNA damage and compromised DNA repair observed in the *PPM1D*-mutant cells, we hypothesized that loss of SOD1 may exacerbate genomic instability, ultimately leading to mutant cell death. To assess this hypothesis, we performed comet assays after *SOD1* deletion. Contrary to our hypothesis, genetic deletion of *SOD1* did not result in a significant increase in DNA breaks in either WT or mutant cells (*Figure 5F*). This suggests that the vulnerability of *PPM1D*-mutant cells to SOD1 loss is not mediated by an exacerbation of DNA damage. Rather, the dependency may be due to other consequences of SOD1 dysregulation, such as altered redox signaling.

## Discussion

The search for synthetic-lethal strategies for cancer therapy has gained significant attention in recent years due to the potential to identify new therapeutic targets that exploit tumor-specific vulnerabilities. In this study, we performed whole-genome CRISPR/Cas9 screening to uncover synthetic-lethal partners of *PPM1D*-mutant leukemia cells. Our screen revealed that *SOD1* was the top essential gene for *PPM1D*-mutant cell survival, a dependency that was validated in vivo. Ongoing efforts are underway to develop SOD1 inhibitors for the treatment of cancer and ALS (*Abati et al., 2020*; *Huang et al., 2000*), and it is conceivable these may be useful in the context of *PPM1D* mutation.

To explore this concept, we tested the sensitivity of WT and *PPM1D*-mutant cells to known SOD1 inhibitors ATN-224 and LCS-1. We found that *PPM1D*-mutant cell lines were significantly more sensitive

to these compounds compared to WT. This sensitivity could be rescued upon supplementation with the antioxidant, NAC, consistent with a role in reducing the impact of ROS. However, given potential off-target effects of LCS-1 (*Ling et al., 2022*; *Steverding and Barcelos, 2020*), we cannot verify that the cytotoxic effects are via its activity toward SOD1. Similarly, we cannot rule out that effects of ATN-224 are not due to other effects caused by copper chelation (*Chidambaram et al., 1984*; *Lee et al., 2013*; *Lowndes et al., 2008*; *Lowndes et al., 2009*). Further work to determine the potential of SOD mimetics like TEMPOL and MnTBAP in mitigating the effects of SOD1 inhibition would be valuable in confirming the specificity of the inhibitors for our underlying phenotype.

We also investigated the mechanisms underlying the dependency on SOD1 and characterized the redox landscape of *PPM1D*-mutant cells, which revealed significant oxidative stress and mitochondrial dysfunction. Recent studies have suggested that PPM1D is indirectly associated with energy metabolism via dephosphorylation of the ATM protein. ATM promotes mitochondrial homeostasis, and therefore sustained inactivation of ATM could lead to potential mitochondrial dysfunction (*Bar et al., 2023*; *Guleria and Chandna, 2016*; *Valentin-Vega et al., 2012*). However, oxidative stress and mitochondrial dysfunction are closely related, and it is difficult to dissect the driving factor. We therefore performed RNA-seq and RPPA analysis to better understand the underlying processes contributing to the heightened oxidative stress observed in the mutant cells. Our analyses indicated a diminished response to oxidative stress in the mutant cells and decreased levels of GSH. These findings may suggest a self-amplifying cycle whereby dysregulation of ROS scavenging systems increases the propensity for oxidative stress, which in turn leads to mitochondrial dysfunction, which further exacerbates oxidative stress. Hence, the additional impairment of ROS detoxification mechanisms within the cell, such as the loss of SOD1, has detrimental consequences for the viability of mutant cells.

The loss of SOD1 leads to increased $O_2^-$ levels and reduced intracellular $H_2O_2$. These two ROS play especially important roles as signaling messengers that control cellular proliferation, differentiation, stress responses, inflammatory responses, and more (*Sauer et al., 2001*; *Sies and Jones, 2020*; *Thannickal and Fanburg, 2000*). These effects are mediated through the reversible oxidation and reduction of cysteine residues (*Poole, 2015*) that have significant effects on key signaling proteins including Erk1/2, protein phosphatases, and more. Therefore, while ROS levels may be significantly impacted by the loss of SOD1, we cannot rule out the possibility of altered ROS-driven signaling, rather than ROS-induced damage, as an underlying mechanism for our results. Follow-up experiments to assess NADPH oxidase and Rac activity may shed further insight on a signaling role for SOD1.

Multiple mechanisms may underlie the suppressed oxidative stress response observed in *PPM1D*-mutant cells. One possible explanation is through PPM1D-mediated inhibition of p53. p53 exhibits complex and context-dependent roles in cellular responses to oxidative stress, and its functions can vary depending on the severity of stress encountered by the cell (*Kang et al., 2013*; *Liang et al., 2013*; *Sablina et al., 2005*). Under mild or moderate oxidative stress conditions, p53 may protect the cell from ROS by inducing the transcription of genes such as SOD, glutathione peroxidase, and others (*Dhar et al., 2011*; *Peuget et al., 2014*; *Sablina et al., 2005*; *Tan et al., 1999*). However, under severe or prolonged oxidative stress, the pro-apoptotic functions of p53 may promote ROS production to eliminate cells that have accumulated excessive DNA damage or irreparable cellular alterations. The duality of these anti- and pro-oxidant functions of p53 highlight its intricate role in modulating responses to oxidative stress. How PPM1D affects the switch between these functions of p53 is not understood. Furthermore, the extent to which the dependency on SOD1 observed in *PPM1D*-mutant cells is mediated through p53 remains unclear and requires deeper exploration to better understand the context in which SOD1 inhibitors can be used in cancer therapy.

Oxidative stress and DNA damage are intimately linked processes that frequently co-occur. Our study also investigated the interplay between PPM1D, DNA damage, and oxidative stress. We demonstrated significant genomic instability of *PPM1D*-mutant cells at baseline and further characterized the effects of mutant PPM1D on specific DNA repair pathways. While previous studies have suggested a role for PPM1D in modulating HR and NHEJ (*Burdova et al., 2019*; *Peng et al., 2015*), our study is the first to demonstrate impaired NHEJ in *PPM1D*-mutant cells. Additionally, our study corroborated previous research demonstrating the synthetic-lethal relationship of *SOD1* and other DNA damage genes such as *RAD54B, BLM,* and *CHEK2* (*Sajesh et al., 2013*; *Sajesh and McManus, 2015*). However, *SOD1* deletion did not exacerbate DNA damage, suggesting that the vulnerability of *PPM1D*-mutant cells to SOD1 loss cannot be explained by increased DNA damage and may be more

likely due to consequences of baseline redox detoxification imbalance or altered redox signaling. Recent studies have shown that ATN-224 can enhance the anti-tumor effects of cisplatin by increasing ROS, decreasing GSH content, and increasing DNA damage (*Li et al., 2022*). These results highlight the potential for combinatorial therapies to achieve therapeutic synergism and underscores the intricate relationship between ROS and DNA damage.

Interestingly, our screen also uncovered sensitivity of *PPM1D*-mutant cells to dropout of genes in the FA DNA repair pathway including *BRIP1* (*FANCJ*), *FANCI*, *FANCA*, *SLX4* (*FANCP*), *UBE2T* (*FANCT*), and *C19orf40* (*FAAP24*). The FA pathway plays a crucial role in facilitating the repair of ICL (*Ceccaldi et al., 2016*; *Kottemann and Smogorzewska, 2013*). Outside of DNA repair and replication, there is a growing body of evidence demonstrating mitochondrial dysfunction and redox imbalance in FA patient cells (*Korkina et al., 1992*). Several FA proteins are implicated in the maintenance of mitochondrial metabolism and mitophagy (*Cappelli et al., 2017*; *Kumari et al., 2014*; *Pagano et al., 2013*; *Sumpter et al., 2016*). Interestingly, a few studies have described a convergence in the FA pathway with SOD1. Early work by Nordenson in 1977 found protective roles for SOD and catalase against spontaneous chromosome breaks in cells from FA patients. Another study demonstrated mitochondrial dysfunction, high ROS levels, and impaired ROS detoxification mechanisms in FA-deficient cell lines (*Kumari et al., 2014*). Interestingly, *SOD1* expression increased in response to $H_2O_2$ treatment in FA-intact cells, but not FA-deficient cells. These findings underscore the critical role of the FA pathway in redox homeostasis by maintaining mitochondrial respiratory function and suppressing intracellular ROS production. Even more importantly, it demonstrates a convergence in the FA pathway with *SOD1*, providing further support for our CRISPR dropout screen results.

In summary, our investigation sheds light on the role of mutant PPM1D in modulating cellular responses to oxidative stress and DNA repair in leukemia cells, offering valuable insights into the underlying molecular mechanisms. This research not only enhances our understanding of PPM1D-mediated cellular responses, but also identifies potential therapeutic targets against *PPM1D*-mutant leukemia cells. However, it is important to acknowledge the limitations of our study. We recognize that while *PPM1D* mutations are frequently observed in patients with t-MN, they are rare in de novo AML (*Hsu et al., 2018*). While there is ample evidence that *PPM1D* is an oncogenic driver in many types of cancers (*Ali et al., 2012*; *Khadka et al., 2022*; *Li et al., 2002*; *Nguyen et al., 2010*; *Wu et al., 2016*), the clinical importance of targeting pre-malignant PPM1D-associated clonal expansion in the hematopoietic system is not clear. However, the prevalence of *PPM1D* somatic mutations in other tissues, such as the esophagus, suggests the need for further investigation (*Yokoyama et al., 2019*).

## Materials and methods
### Cell lines and reagents
The following cell lines were purchased from DSMZ with the catalogue numbers as follows: MOLM-13 (Cat #ACC-554), OCI-AML2 (Cat #ACC-99), and OCI-AML3 (Cat #ACC-582). Heterozygous *PPM1D*-mutant cell lines were previously generated in our lab using CRISPR/Cas9 and used in a previous publication (*Hsu et al., 2018*, PMID: 30388424). All cell lines tested negative for mycoplasma using a PCR-based method. The OCI-AML2, OCI-AML3, MOLM13, and U2OS lines were obtained relatively recently from their original source or through ATCC where they were authenticated or authenticated in our lab. The sex of the cell lines is as follows: MOLM13 – male, OCI-AML2 – male, OCI-AML3 – male, U2-OS – female.

Cas9-expressing OCI-AML2 cells were generated by lentiviral transduction using pKLV2-EF1aBsd2ACas9-W plasmid obtained from Dr. Kosuke Yusa from the Sanger Institute (Addgene #67978). Four days post-transduction, cells underwent blasticidin selection. Single clones were obtained by fluorescence-activated cell sorting and functionally tested for Cas9 activity using a lentiviral reporter pKLV2-U6gRNA5(gGFP)-PGKBFP2AGFP-W (Addgene #67980). *PPM1D*-mutant cell lines were generated using the RNP-based CRISPR/Cas9 delivery method using a single sgRNA (GCTA AAGCCCTGACTTTA). Single cells were sorted into 96-well, round-bottom plates and expanded. Clones were validated by Sanger sequencing, TIDE analysis, and western blot to visualize the overexpressed, truncated mutant protein. Two validated *PPM1D*-mutant clones were selected for the CRISPR dropout screen.

## CRISPR dropout screen and analyses

For large-scale production of lentivirus, 15 cm plates of 80–90% confluent 293T cells were transfected using Lipofectamine 2000 (Invitrogen) with 7.5 µg of the Human Improved Whole-Genome Knockout CRISPR library V1 (by Kosuke Yuya, Addgene #67989), 18.5 µg of psPax2, and 4 µg of pMD2.G. A lentivirus titer curve was performed prior to the screen to determine the volume of viral supernatant to add for a multiplicity of infection of ~0.3. For the CRISPR dropout screen, one WT and two independent *PPM1D*-mutant Cas9-expressing OCI-AML2 cell lines were used as biological replicates, with three technical replicates per line. $3 \times 10^7$ cells were transduced with the lentivirus library supernatant. Three days post-transduction, the cells were selected with puromycin for 3 days. Cells were collected on day 28 for genomic DNA isolation using isopropanol precipitation. Illumina adapters and barcodes were added to samples by PCR as previously described (*Tzelepis et al., 2016*). Single-end sequencing was performed on the HiSeq 2000 V4 platform and cell-essential genes were identified using the MaGECK-VISPR (*Li et al., 2014*).

## Competitive proliferation assay

Gene-specific sgRNAs were cloned into the pKLV2-U6gRNA5(BbsI)-PGKpuro2ABFP (Addgene #67974) lentiviral backbone. 293T cells ($0.4 \times 10^6$ cells/well) were seeded in a six-well plate the day prior and transfected using Lipofectamine 3000 with pMD2G (0.8 µg), pAX2 (1.6 µg), and the sgRNA-BFP (1.6 µg) plasmids. Cas9-expressing cells were then seeded in 12-well plates (200k cells/well, in triplicates) in media supplemented with 8 µg/mL polybrene and 5 µg/mL blasticidin, and lentivirally transduced at a titer that yields 50% infection efficiency. Cells were assayed using flow cytometry for BFP expression between 4 and 16 days post-transduction and normalized to the BFP percentage at day 4.

## Drug and proliferation assays

Drug and proliferation assays were done using the Cell Proliferation MTT Kit (Sigma) as per the manufacturer's protocol. Briefly, $1 \times 10^4$ cells were plated in 96-well, flat-bottom plates and treated with vehicle or drugs in a total volume of 100 µL. Plates were incubated at 37°C for at least 24 hr. 10 µL of MTT labeling reagent was added to each well and incubated for 4 hr. 100 µL of solubilization buffer was added to each well and incubated overnight. Plates were analyzed using a fluorometric microplate reader at 550 nm. Stock solutions of ATN-224 (Cayman Chemical #23553) and LCS-1 (MedChem, HY-115445) were in DMSO and frozen in –20°C.

## SOD activity assay

SOD activity was measured per the manufacturer's protocol (Invitrogen, Cat#EIASODC). Briefly, cells were treated with low- or high-dose ATN-224 (6.25 µM and 12.5 µM, respectively) for 16 hr and harvested. Cells were washed with PBS and lysed with ice-cold NP-40 lysis buffer (Invitrogen #FNN0021) with protease inhibitor (Thermo Fisher, #78440). Cells were sonicated for 5 s × 5 rounds and then spun at 13,000 rpm for 10 min at 4°C. Protein concentrations were measured using BCA assay (Thermo Fisher, #23225) and diluted to a concentration of 10 µg/µL. 100 µg (10 µL) of protein was loaded per sample and incubated for 20 min with the added substrates. Plates were read on a microplate reader at 450 nm.

## Intravenous transplantation of leukemia cells in NSG mice

WT and PPM1D-mutant OCI-AML2 cells were transduced with EV or sgSOD1 lentivirus, as described in the 'Competitive proliferation assay' section above. Three days post-transduction, cells underwent puromycin selection (3 µg/mL). On day 6 post-transduction, the infection rate was determined by flow cytometry using the percentage of BFP+ cells. All samples had an infection rate of >95%. 8-week-old male NOD.Cg-*Prkdc*$^{scid}$*Il2rg*$^{tm1Wjl}$/SzJ (NSG) mice were purchased from The Jackson Laboratory (strain #005557) and sublethally irradiated (250 cGy) immediately prior to transplantation. $2 \times 10^6$ cells were intravenously injected in the tail vein of mice (n=8 per group). After transplantation, mice were monitored daily for disease progression and humane euthanasia was performed when animals lost >15% body weight or had signs of severe disease (limb paralysis, decreased activity, and hunching). All animal procedures and studies were done in accordance with the Institutional Animal Care and Use Committee (IACUC).

## Alkaline comet assay

Comet assays were conducted as previously described (*Greve et al., 2012*; *Schmezer et al., 2001*). Cells were resuspended to $1 \times 10^5$ cells/mL and mixed with 1% low-melting agarose (R&D Systems) at a 1:10 ratio and plated on two-well comet slides (R&D Systems). Cells were then lysed overnight and immersed in alkaline unwinding solution as per the manufacturer's protocol (Trevigen). Fluorescence microscopy was performed at ×10 magnification using the Keyence BZ-X800 microscope and analyses of comet tails were performed using the Comet Assay IV software (Instem). At least 150 comet tails were measured per sample.

## Chromosome aberration analysis of mitotic chromosome spreads

Primary resting mouse splenic B-cells were isolated using anti-CD43 microbeads (Miltenyi Biotec) and activated with 25 µg/mL LPS (Sigma), 5 ng/mL IL-4 (Sigma), and 0.5 µg/mL anti-CD180 (BD Pharmingen) for 30 hr. The cells were then treated with cisplatin for 16 hr at two concentrations – 0.5 µM and 5 µM cisplatin. Metaphases were prepared as previously described (*Zong et al., 2019*). Briefly, cells were arrested at mitosis with colcemid (0.1 µg/mL, Thermo Fisher) for 1 hr. Cells were then incubated in a prewarmed, hypotonic solution of potassium chloride (75 mM) for 20 min to induce swelling and fixed in methanol/glacial acetic acid (3:1). Droplets were spread onto glass slides inside a cytogenetic drying chamber. Fluorescence in situ hybridization was performed using a Cy3-labeled peptide nucleic acid probe to stain telomeres and DNA was counterstained by DAPI. At least 50 metaphases were scored for chromosome aberrations for each experimental group.

## ROS assays

To measure superoxide, total cellular ROS, and lipid peroxidation, $1 \times 10^6$ cells were collected after the indicated treatments and washed with PBS. The cells were stained with 1 µM MitoSOX Green (Thermo Fisher), 5 µM dihydroethidium (Thermo Fisher), 20 µM DCFDA (Abcam), or 2.5 µM BODIPY 581/591 (Thermo Fisher) in FBS-free Hanks' buffered saline solution (Thermo Fisher), and incubated at 37°C for 30 min. The staining was quenched with flow buffer (PBS, 2% FBS, 1% HEPES) and washed twice before resuspension in DAPI-containing flow buffer to assess ROS in viable cells. For detection of intracellular GSH, we utilized the Intracellular GSH Detection Assay Kit (Abcam) as per the manufacturer's protocol. The data was acquired using an LSRII (BD Biosciences) and analyzed on FlowJo. The mean fluorescence intensity was used for data analysis.

## Reverse-phase protein array

RPPA assays for antibodies to proteins or phosphorylated proteins in different functional pathways were carried out as described previously (*Coarfa et al., 2021*; *Lu et al., 2021*; *Wang et al., 2022*). Specifically, protein lysates were prepared from cultured cells with modified Tissue Protein Extraction Reagent (TPER) (Life Technologies Corporation, Carlsbad, CA, USA) and a cocktail of protease and phosphatase inhibitors (Roche, Pleasanton, CA, USA) (*Lu et al., 2021*). The lysates were diluted into 0.5 mg/mL in SDS sample buffer and denatured on the same day. The Quanterix 2470 Arrayer (Quanterix, Billerica, MA, USA) with a 40 pin (185 µm) configuration was used to spot samples and control lysates onto nitrocellulose-coated slides (Grace Bio-Labs, Bend, OR, USA) using an array format of 960 lysates/slide (2880 spots/slide). The slides were processed as described and probed with a set of 264 antibodies against total proteins and phosphoproteins using an automated slide stainer Autolink 48 (Dako, Santa Clara, CA, USA). Each slide was incubated with one specific primary antibody and a negative control slide was incubated with antibody diluent without any primary antibody. Primary antibody binding was detected using a biotinylated secondary antibody followed by streptavidin-conjugated IRDye680 fluorophore (LI-COR Biosciences, Lincoln, NE, USA). Total protein content of each spotted lysate was assessed by fluorescent staining with Sypro Ruby Protein Blot Stain according to the manufacturer's instructions (Molecular Probes, Eugene, OR, USA).

Fluorescence-labeled slides were scanned on a GenePix 4400 AL scanner, along with accompanying negative control slides, at an appropriate PMT to obtain optimal signal for this specific set of samples. The images were analyzed with GenePix Pro 7.0 (Molecular Devices, Silicon Valley, CA, USA). Total fluorescence signal intensities of each spot were obtained after subtraction of the local background signal for each slide and were then normalized for variation in total protein, background, and non-specific labeling using a group-based normalization method as described (*Lu et al., 2021*).

For each spot on the array, the background-subtracted foreground signal intensity was subtracted by the corresponding signal intensity of the negative control slide (omission of primary antibody) and then normalized to the corresponding signal intensity of total protein for that spot. Each image, along with its normalized data, was evaluated for quality through manual inspection and control samples. Antibody slides that failed the quality inspection were either repeated at the end of the staining runs or removed before data reporting. A total of 261 antibodies remained in the list. Multiple t-tests with Benjamini-Hochberg correction were performed for statistical analysis and filtering was based on an FDR < 0.2 and linear fold change of >1.25.

## RNA-seq

Bulk RNA-seq was performed on WT and *PPM1D*-mutant OCI-AML2 cells after lentiviral *SOD1* CRISPR knockout. Cells were transduced with pKLV2-U6-sgRNA-BFP lentivirus (either EV or with *SOD1*-sgRNA). Transduced cells were then cultured for 10 days and BFP+ cells were sorted directly into Buffer RLT Plus with β-mercaptoethanol. RNA was isolated using the Allprep DNA/RNA Micro Kit (QIAGEN) per the manufacturer's protocols. RNA-seq library preparation was done using the True-Seq Stranded mRNA kit (Illumina) per the manufacturer's protocol. Quality control of libraries was performed using a TapeStation D1000 ScreenTape (Agilent, 5067-5584). Libraries were then sequenced using an Illumina NextSeq 2000 sequencer, aiming for >20 million reads per biological replicate. Paired-end RNA-seq reads were obtained and trimmed using trimGalore (https://github.com/FelixKrueger/TrimGalore; *Krueger, 2023*). Mapping was performed using the STAR package (*Dobin et al., 2013*) against the human genome build UCSC hg38 and counts were quantified with featureCounts (*Liao et al., 2014*). Differential expression analysis was performed using the DESeq2 R package (1.28.1) (*Love et al., 2014*). p-Values were adjusted with Benjamini and Hochberg's approach for controlling the FDR. Significant differentially expressed genes between the indicated comparisons were filtered based on an FDR < 0.05 and absolute fold change exceeding 1.5. Pathway enrichment analysis was carried out using the GSEA (http://software.broadinstitute.org/gsea/index.jsp) software package and significance was achieved for adjusted FDR < 0.25.

## Seahorse assay

Mitochondrial bioenergetics in AML cell lines were performed using the Seahorse XFp Cell Mito Stress Kit (Agilent Technologies) on the Seahorse XFe96 Analyzer. Cells were resuspended in XF RPMI base media supplemented with 1 mM pyruvate, 2 mM L-glutamine, 10 mM glucose. $1 \times 10^5$ cells/well were seeded in poly-D-lysine (Thermo Fisher) coated XFe96 plates. The plate was incubated in a non-$CO_2$ incubator at 37°C for 1 hr to equilibrate. OCR and ECAR measurements were taken at baseline and every 8 min after sequential addition of oligomycin (2 µM), FCCP (0.5 µM), and rotenone/antimycin A (0.75 µM). All measurements were normalized to the number of viable cells.

## Generation of *PPM1D* WT U2OS cells using CRISPR editing

U2OS cells containing the DR-GFP (for HR) or EJ5-GFP (for NHEJ) DNA repair reporter cassettes were kindly provided by the Bertuch Lab at Baylor College of Medicine. To establish *PPM1D*-WT isogenic lines, knock-in CRISPR editing was performed with a single-stranded oligodeoxynucleotide (ssODN) template: TGCCCTGGTTC GTAGCAATGCCTTCTCAGAGAATTTTCTAGAGGTTTCAGCTGAGATAG CTCGTGAGAATGTACAAGGTGTAGTCATACCCTAAAAGATCCAGAACCACTTGAAGAAAATGCG CTAAAGCCCTGACTTTAAGGATACA. The *PPM1D* sgRNA sequence used was: ATAGCTCGAGA GAATGTCCA. 1.3 µg of Cas9 (PNA Bio) was incubated with 1 µg of sgRNA for 15 min at room temperature. 1 µg of the ssODN template was then added to the Cas9-sgRNA complexes and mixed with 20,000 U2OS cells and resuspended in 10 µL of Buffer R, immediately prior to electroporation. The neon electroporation system was used with the following conditions: 1400 V, 15 ms, 4 pulses. Single cell-derived clones were genotyped by Sanger sequencing and PPM1D protein expression was validated by western blot.

## GFP reporter-based DNA repair assays

For the DNA repair reporter assay, 100,000 U2OS cells were seeded in a 12-well plate in antibiotic-free Dulbecco's Modified Eagle Medium (Thermo Fisher) supplemented with 10% FBS. Cells were transfected with 3.6 µL of Lipofectamine 2000 (Invitrogen) in 200 µL of OptiMEM with 0.8 µg of the

I-SceI expression plasmid (pCBASce, Addgene #60960). The media was replaced the next morning and the cells were trypsinized 48 hr post-transfection for analysis of GFP expression by flow cytometry (BD Biosciences).

## Immunofluorescence microscopy

12 mm glass coverslips were coated with 50 µg/mL poly-D-lysine (Thermo Fisher) for 30 min at room temperature and washed with sterile PBS. $0.5 \times 10^6$ suspension cells/well were seeded on coverslips and incubated for 1 hr at 37°C to allow for adherence. Samples were then fixed with 4% paraformaldehyde for 10 min at 37°C and washed three times with 0.01% Triton-X PBS (PBS-T). Fixed cells were permeabilized with 0.5% PBS-T for 20 min, washed three times, and incubated with 5% goat serum (Thermo Fisher) for 1 hr at room temperature. Afterward, samples were incubated overnight at 4°C with the following primary antibodies: rabbit anti-Rad51 (Cell Signaling #8875S 1:100) or rabbit anti-53BP1 (Thermo Fisher #PA1-16565, 1:500). The following day, samples were washed and incubated at room temperature for 1 hr with Alexa Fluor 488-conjugated goat anti-rabbit IgG (#111-545-144, Jackson ImmunoResearch, 1:500). After secondary antibody incubation, the coverslips were washed three times with PBS and mounted with fluoromount-G mounting medium with DAPI (Thermo Fisher) on glass microscope slides and sealed with nail polish. Imaging was done on the Keyence BZ-X800 microscope and foci analysis was performed using CellProfiler.

## Immunoblotting

Cells were lysed with 1× RIPA buffer supplemented with Halt Protease and Phosphatase inhibitor cocktail (Thermo Fisher) for 1 hr at 4°C. Protein concentration was quantified using the Pierce BCA protein assay kit (Thermo Fisher) and boiled at 95°C in 1× Laemmli (Bio-Rad) for 7 min. The samples in which mitochondrial proteins were probed were not boiled, as boiling can cause signal reduction. Instead, samples were warmed to 37°C for 30 min prior to loading. The proteins were separated by SDS-PAGE on 4–15% gradient gels (Bio-Rad) and transferred onto PVDF membranes using the iBlot Dry Blotting system (Thermo Fisher). Membranes were incubated for 1 hr at room temperature in 5% milk in Tris-buffered saline solution with Tween-20 (TBST). After washing, the membranes were incubated overnight at 4°C with the following primary antibodies: mouse anti-PPM1D (F-10, Santa Cruz, 1:1000), mouse anti-GAPDH (MAB374, Millipore, 1:200), mouse total OXPHOS Human antibody cocktail (ab110411, Abcam, 1:1000), mouse anti-vinculin (V9131, Sigma-Aldrich, 1:2000). The following day, membranes were washed twice with TBST and incubated for 1 hr with HRP-linked anti-rabbit IgG or anti-mouse IgG (Cell Signaling, 1:5000–1:10,000) at room temperature. Blots were imaged on the Bio-Rad ChemiDoc platform.

## Statistical analysis

Statistical analysis incorporated in the MaGECK-VISPR algorithm includes p-value and FDR calculations. GraphPad Prism 6.0 was used for other statistical analyses. The sample size (n) specified in the Figure Legends was used for statistical analysis and denotes the number of independent biological replicates. The main conclusions were supported by data obtained from at least two biological replicates. The graphs presented in the figures are shown with error bars indicating either mean ± SEM or mean ± SD, as mentioned in the Figure Legends. Two-tailed t-tests were performed to calculate statistics, assuming unequal standard deviations, unless mentioned otherwise. Significance levels are indicated in the figures and were determined using GraphPad Prism. Results were considered statistically significant at *$p < 0.05$, **$p < 0.01$, ***$p < 0.001$, ****$p < 0.0001$.

## Acknowledgements

This work was supported by R01CA237291 and P01CA265748. This work was also supported by the NCI Cancer Center Support Grant P30CA125123 which partly supports the Cytometry Core, the Proteomics & Metabolomics Core, and the Antibody-based Proteomics Core. Support for the cores was also provided by the Cancer Prevention and Research Institute of Texas (CPRIT) from grants: RP180672, RR024574, and RP210227 and NIH S10OD028648. LZ was supported by the Baylor Research Advocates for Student Scientists (BRASS) Foundation and the Janice McNair Medical Foundation.

## Additional information

### Funding

| Funder | Grant reference number | Author |
| --- | --- | --- |
| National Cancer Institute | R01CA237291 | Linda Zhang<br>Joanne I Hsu<br>Chun-Wei Chen<br>Alejandra G Martell<br>Anna G Guzman<br>Katharina Wohlan<br>Sarah M Waldvogel<br>Ayala Tovy<br>Margaret A Goodell |
| National Cancer Institute | P01CA265748 | Linda Zhang<br>Chun-Wei Chen<br>Alejandra G Martell<br>Anna G Guzman<br>Katharina Wohlan<br>Sarah M Waldvogel<br>Margaret A Goodell |
| National Institute of Diabetes and Digestive and Kidney Diseases | F30DK116428 | Joanne I Hsu |
| Eunice Kennedy Shriver National Institute of Child Health & Human Development | F30HD111129 | Sarah M Waldvogel |
| Leukemia and Lymphoma Society | Scholar Award | Hidetaka Uryu<br>Koichi Takahashi |
| Baylor Research Advocates for Student Scientists (BRASS) Foundation | | Linda Zhang |
| McNair Foundation | | Linda Zhang<br>Sarah M Waldvogel |
| National Cancer Institute | P30CA125123 | Shixia Huang<br>Cristian Coarfa |

The funders had no role in study design, data collection and interpretation, or the decision to submit the work for publication.

### Author contributions

Linda Zhang, Conceptualization, Resources, Data curation, Software, Formal analysis, Funding acquisition, Validation, Investigation, Visualization, Methodology, Writing – original draft, Project administration, Writing – review and editing; Joanne I Hsu, Conceptualization, Resources, Data curation, Investigation, Methodology, Writing – review and editing; Etienne D Braekeleer, Conceptualization, Data curation, Formal analysis, Methodology, Writing – review and editing; Chun-Wei Chen, Resources, Software, Formal analysis, Investigation, Methodology, Writing – review and editing; Tajhal D Patel, Formal analysis, Visualization, Methodology, Writing – original draft, Writing – review and editing; Alejandra G Martell, Data curation, Formal analysis; Anna G Guzman, Data curation, Funding acquisition, Investigation; Katharina Wohlan, Data curation, Investigation, Methodology; Sarah M Waldvogel, Conceptualization, Investigation, Methodology, Writing – review and editing; Hidetaka Uryu, Data curation, Formal analysis, Investigation, Methodology; Ayala Tovy, Conceptualization, Data curation, Methodology, Writing – review and editing; Elsa Callen, Data curation, Formal analysis, Methodology, Writing – review and editing; Rebecca L Murdaugh, Resources, Data curation, Formal analysis, Investigation; Rosemary Richard, Resources, Data curation, Methodology; Sandra Jansen, Lisenka Vissers, Bert BA de Vries, Resources; Andre Nussenzweig, Resources, Data curation, Formal analysis, Funding acquisition, Methodology; Shixia Huang, Resources, Data curation, Formal analysis, Investigation, Writing – original draft, Writing – review and editing; Cristian Coarfa, Formal analysis, Methodology; Jamie Anastas, Data curation, Formal analysis, Supervision, Methodology; Koichi

Takahashi, Resources, Data curation, Formal analysis, Investigation, Methodology; George Vassiliou, Conceptualization, Resources, Data curation, Formal analysis, Methodology, Writing – review and editing; Margaret A Goodell, Conceptualization, Resources, Software, Supervision, Funding acquisition, Investigation, Visualization, Methodology, Writing – original draft, Project administration, Writing – review and editing

## Author ORCIDs
Linda Zhang ⬤ http://orcid.org/0000-0002-5777-2357
Hidetaka Uryu ⬤ http://orcid.org/0000-0001-7167-895X
George Vassiliou ⬤ http://orcid.org/0000-0003-4337-8022
Margaret A Goodell ⬤ http://orcid.org/0000-0003-1111-2932

## Ethics

This study was performed in accordance with the recommendations in the Guide for the Care and Use of Laboratory Animals in the National Institutes of Health. All animals were housed in AAALAC-accredited, specific-pathogen-free animal care facilities at Baylor College of Medicine (BCM) and all procedures were approved by the BCM Institutional Animal Care and Use Committee (IACUC) (protocol #AN-2234). Mice of both sexes were used, and experimental mice were separated by sex and housed with 4 mice per cage. All mice were immune-competent and healthy prior to experiments described. All procedures were performed under isoflurane anesthesia and every effort was made to minimize animal suffering.

Reviewer #1 (Public Review): https://doi.org/10.7554/eLife.91611.3.sa1
Reviewer #2 (Public Review): https://doi.org/10.7554/eLife.91611.3.sa2
Reviewer #3 (Public Review): https://doi.org/10.7554/eLife.91611.3.sa3
Author response https://doi.org/10.7554/eLife.91611.3.sa4

# Additional files

## Supplementary files
• MDAR checklist

## Data availability

All raw and processed sequencing data generated in this work is publicly available at GEO data repository under the accession number GSE240874. All data generated and analyzed during the study, including imaging and western blot files, are included in the manuscript and supporting files. Source data files have been provided for *Figures 1, 3–5*, *Figure 1—figure supplement 1*, *Figure 2—figure supplement 2*, *Figure 5—figure supplement 1*. Source data for the dropout screen, RNA-seq, and RPPA analyses have also been provided for *Figures 1 and 4*. Any additional information required to reanalyze the data reported in this paper is available from the lead contact upon request. The human cell lines generated by the Goodell laboratory for this study are available upon request and will require a standard Materials Transfer Agreement (MTA).

The following dataset was generated:

| Author(s) | Year | Dataset title | Dataset URL | Database and Identifier |
|---|---|---|---|---|
| Zhang L, Hsu JI, Braekeleer ED, Chen CW, Patel TD, Urya H, Guzman AG, Martell AM, Waldvogel SM, Tovy A, Callen E, Murdaugh R, Richard R, Jansen S, Vissers L, de Vries BA, Nussenzweig A, Huang SX, Coarfa C, Anastas JN, Takahashi K, Vassiliou G, Goodell MA | 2024 | SOD1 is a synthetic lethal target in PPM1D-mutant leukemia cells | https://www.ncbi.nlm.nih.gov/geo/query/acc.cgi?acc=GSE240874 | NCBI Gene Expression Omnibus, GSE240874 |

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
