## [Editor Report · eLife assessment]

Gain-of-function mutations and amplifications of PPM1D are found across several human cancers and are associated with advanced tumor stage and worse prognosis. Thus far, the clinical translation has not been possible due to the lack of PPM1D inhibitors with favorable pharmacokinetic properties. This **useful** study leverages CRISPR/Cas9 screening to determine that loss of SOD1 and is synthetic lethal with PPM1D mutation in leukemia. The mechanistic analyses are still **incomplete**.

---

## [Referee Report · Reviewer #1 (Public Review)]

Summary:

Gain-of-function mutations and amplifications of PPM1D are fond across several human cancers and are associated with advanced tumor stage, worse prognosis, and increased lymph node metastasis. This manuscript presents important findings that SOD1 inhibition is a potential strategy to achieve therapeutic synergism for PPM1D-mutant leukemia; and demonstrates the redox landscape of PPM1D-mutant cells.

Strengths:

In this manuscript, Zhang and colleagues investigate the synthetic-lethal dependencies of PPM1D (protein phosphatase, Mg2+/Mn2+ dependent 1D) in leukemia cells using CRISPR/Cas9 screening. They identified that SOD1 (superoxide dismutase-1) as the top hit, whose loss reduces cellular growth in PPM1D-mutant cells, but not wildtype (WT) cells. Consistently, the authors demonstrate that PPM1D-mutant cells are more sensitive to SOD1 inhibitor treatment. By performing different in vitro studies, they show that PPM1D-mutant leukemia cells have elevated level of reactive oxygen species (ROS), decreased basal respiration, increased genomic instability, and impaired non-homologous end-joining repair. These data highlight the potential of SOD1 inhibition as a strategy to achieve therapeutic synergism for PPM1D-mutant leukemia; and demonstrates the redox landscape of PPM1D-mutant cells.

Weaknesses:

While the current study has identified synthetic lethality of PPM1D-mutant leukemia cells upon SOD1 inhibition, the underlying mechanism remains elusive. Although ROS levels have been assessed between wild-type (WT) and PPM1D-mutant leukemia cells, the specific redox alterations induced by SOD1 inhibition in PPM1D mutant versus WT cells have not been elucidated. To address this gap, direct comparisons of ROS levels using various probes should be conducted between PPM1D mutant and WT cells under conditions of SOD1 inhibition.

---

## [Referee Report · Reviewer #2 (Public Review)]

The authors used a whole genome CRISPR screen to identify targetable synthetic lethalities associated with PPM1D mutations, known poor prognosis and currently undruggable factors in leukemia. The authors identified the cytosolic superoxide dismutase (SOD1, Cu/Zn SOD) as a major protective factor in PPMD1 mutant vs. wt cells, and their study investigates associated mechanisms of this protection. Using both genetic depletion and small molecule inhibitors of SOD1, the authors conclude that SOD1 loss exacerbates mitochondrial dysfunction, ROS levels and DNA damage phenotypes in PPM1D mutant cells, decreasing cell growth in AML cells. The data strongly support that PPMD1 mutant cells have high levels of total peroxides and elevated DNA breaks, and that genetic depletion of SOD1 decreases cell growth in two AML cell lines. However, the authors don't explain how superoxide radical (which is not damaging by itself) induces such damage, the on-target effects of the SOD1 inhibitors at the concentrations is not clear, the increase in total hydroperoxides is not supported by loss of SOD1, the changes in mitochondrial function are small, and there is no assessment of how the mitochondrial SOD2 expression or function, which dismutates mitochondrial superoxide, is altered. Overall these studies do not distinguish between signal vs. damaging aspects of ROS in their models and do not rule out an alternate hypothesis that loss of SOD1 increases superoxide production by cytosolic NADPH activity which would significantly alter ROS-driven regulation of kinase/phosphatase signal modulation, affecting cell growth and proliferation as well as DNA repair. Additionally, with the exception of growth defects demonstrated with sgSOD1, the majority of data are acquired using two chemical inhibitors, LCS1 and ATN-224, without supporting evidence that these inhibitors are acting in an on-target manner.

Overall, the authors address an important problem by seeking targetable vulnerabilities in PPM1D mutant AML cells, it is clear SOD1 deletion induces strong growth defects in the AML cell lines tested, most of the approaches are appropriate for the outcomes being evaluated, and the data are technically solid and well-presented. The major weakness lies in which redox pathways and ROS species are evaluated, how the resulting data are interpreted, and gaps in the follow-up experiments. Due to these omissions, as currently presented, the broader impact of these findings are unclear.

These specific concerns are outlined in detail below and I offer some suggestions regarding how to clarify the mechanisms underlying their initial observation of SOD1 synthetic lethality:

(1) Fig. 1 - SOD1 appears to be clustered with several other genes in the volcano plot (including FANC proteins). Did any other ROS-detoxifying enzymes show similar fitness scores? The effects of the SOD1 sgRNA are striking, however it would be useful to see qPCR or immunoblot data confirming robust depletion.

Does SOD1 co-expression in PPM1-mutant patient AML correspond to poorer disease outcomes? This can be evaluated in publicly available patient datasets and would support the idea of SOD1 synthetic lethality.

It would also be useful to know (given the subsequent results) whether expression of the SOD2, the mitochondrial superoxide dismutase, is altered in response to SOD1 loss.

(2) Fig. 2 - What are the relative SOD1 levels in the mutant PPM1D vs. wt. cell lines? The effects of the chemical inhibitors are stronger in MOLM-13 than the other two lines. These data could also point to whether LCS-1 and ATN-224 cytotoxicity is on-target or off-target at these concentrations, which is a key issue not currently addressed in these studies. This is a particular concern as the OCI-AML2 line shows a stronger growth defect with CRISPR SOD1 KO (in Fig 1) but the smallest effects with these chemical inhibitors.

While endogenous mitochondrial superoxide levels are elevated in PPM1D mutant lines, it is entirely unclear why SOD1 inhibition should affect mitochondrial superoxide as it detoxifies cytosolic superoxide. Also unclear why DCFDA signal (which measures total hydroperoxides) is *increased* under SOD1 inhibition - SOD1 dismutates superoxide radicals into hydrogen peroxide, therefore unless SOD2 is compensating for SOD1 loss, one might expect hydroperoxides to be lower (unless some entirely different oxidase is increasing their levels). None of these outcomes appear to be considered. Finally, it is not explained how lipid peroxidation, which requires production of hydroxyl or similarly high potency radicals, is being caused by increased superoxide or peroxides. One possibility is there is an increase in labile iron, in which case this phenotype would be rescued by the iron chelator desferal, and by the lipophilic antioxidant, ferrostatin.

Do the sgSOD1 cells also show similar increases in MitoSox green, DCFDA and BODIPY signal? These experiments would clarify whether the effects with the inhibitors are directly related directly to SOD1 loss or if they represent off-target effects from the inhibitors and/or compensatory changes in SOD2.

(3) Fig. 3 - the effects on mitochondrial respiratory parameters, while statistically significant, do not seem biologically striking. Also, these data are shown for OCI-AML2 cells which show the smallest cytotoxic effects with the SOD1 inhibitors among the 3 lines tested. They do however show the most robust growth defect with sgSOD1. This discrepancy could suggest that mitochondrial dysfunction does not underlie the observed growth defect and/or the inhibitor cytotoxicity is not on-target. Ideally mitochondrial profiling should also be carried out on this cell line with inducible SOD1 depletion. Have the authors assessed whether the mitochondrial Bcl family proteins are affected by the inhibitors?

(4) Fig. 4 - Currently the data in this figure do not support the authors claim that PPM1D-mutant cells have impaired antioxidant defense mechanisms, leading to an elevation in ROS levels and reliance on SOD1 for protection. It should be noted that oxidative stress specifically refers to adverse cellular effects of increasing ROS, not baseline levels of various redox parameters. Ideally levels of GSSG/GSH would be a better measure of potential redox stress tolerance than the total antioxidant capacity assay. Finally, oxidative stress can be assessed by challenging the wt and mutant PPM1D cell lines with oxidant stressors such as paraquat which elevates superoxide or drugs like erastin which elevate mitochondrial ROS. The immunoblot shows negligible changes in the antioxidant proteins assayed. Again, this blot should include SOD2 which is the most relevant antioxidant in the context of mitochondrial superoxide.

(5) Fig. 5 - These data support that DNA breaks are elevated in PPM1D mutant vs. wt cells. However, the data with the chemical SOD1 inhibitor again do not convince that the enhanced levels are due to on-target effects on SOD1. Use of the alkaline comet assay is appropriate for these studies and the 8-oxoguanine data do indicate contributions from oxidative DNA base damage. But these are unlikely to result directly from altered superoxide levels, as this species cannot directly oxidize DNA bases or cause DNA strand breaks.

The following points summarize my specific experimental and textual recommendations:

(1) These studies require an assessment of on-target efficacy of the inhibitors at the relevant concentration ranges. Ideally, they should have minimal effects against SOD1 knockout cell lines (acute challenge at a time point before the growth defects become apparent) and show better efficacy in SOD1-overexpressing lines. Key experiments (changes in superoxide, OCR profiling, DNA alkaline comet assay) would be more convincing if they are carried out with SOD1 knockout lines to compare against the inhibitor effects (3-4 days after introducing sgSOD1 when growth defects are not apparent).

(2) Instead of using NAC, which elevates glutathione synthesis but also has several known side-effects, the authors may want to determine whether Tempol, a SOD mimetic can rescue the effects of SOD1 knockout or inhibition. This would directly prove that SOD1 functional loss underlies the observed growth defect and cytotoxicity from genetic SOD1 knockdown or chemical inhibition.

(3) The complete lack of consideration of SOD2 in these studies is a missed opportunity as it reduces mitochondrial superoxide levels but elevates hydrogen peroxide levels. It would be very interesting to see whether SOD1 inhibition leads to compensatory increases in SOD2. SOD2 can be easily measured by immunoblot. Furthermore, measuring total superoxide via hydroethidium in a flow cytometric assay vs. mitochondrial ROS in PPM1D mut vs. wt cells and under SOD1 knockout would enable a determination of which species dominates (cytosolic or mitochondrial). These experiments are required to fill some logical gaps in interpretation of their redox data.

(4) Given the DNA breaks observed in PPM1D mutant cells, it is highly recommended the authors assess whether iron levels are elevated in mut vs. wt cells and whether desferal can rescue observed SOD1 inhibition defects.

(5) The authors may want to assess whether Rac1 or NADPH oxidase activity is altered in the SOD1 KO in wt vs. PPM1D cells. Their results may be the consequence of compromised ROS-driven survival signaling or DNA repair rather than direct ROS-induced damage, which is not caused directly by superoxide (or hydrogen peroxide).

(6) It is recommended the discussion focus more strongly on how the signaling function of superoxide vs. its reactions with other molecular entities to induce genotoxic outcomes could be contributing to the observed phenotypes. The discussion of FANC proteins, which were targets with similar fitness scores but not experimentally investigated at all, is an unwarranted digression.

---

## [Referee Report · Reviewer #3 (Public Review)]

Summary:

Authors performed a genome-wide CRISPR-based screen for synthetic lethal interactions in leukemic cells expressing a mutant form of PPM1D and identified SOD1. Loss of SOD1 or its inhibition with small molecule compounds reduced survival of the cells containing truncated PPM1D. Further analysis revealed that mitochondria are functionally deficient in PPM1D mutant cells resulting in increased levels of ROS. Surprisingly, expression profiling and reverse phase protein arrays revealed that PPM1D mutant cells did not respond appropriately to the increased levels of ROS. The precise molecular mechanism underlying this phenotype remains currently unclear, nevertheless the study convincingly shows that PPM1D mutant cells are vulnerable to oxidative stress.

Strengths:

Experimental procedures used in the study are appropriate and overall the presented data are very convincing. The study identified an important vulnerability of leukemic cells that carry PPM1D mutation and provides a fundamental background for testing SOD1 inhibitors in preclinical research. In the revised version of the manuscript, authors provide several new experiments that support their former conclusions. In particular, they showed that deletion of SOD1 in AML cells improved survival of the transplanted mice and this effect was more prominent when using cells carrying the mutant PPM1D. Further, they included an important control experiment that showed decreased SOD1 activity after treatment with ATN-224 inhibitor.

Weaknesses:

In the opinion of reviewer, there are no obvious weaknesses in this study. In broader view, the findings presented here using in vitro cultures will need to be validated in vivo by future research. Cell lines used in the study were generated by CRSIPR approaches in AML cells that have already been transformed. In addition, genome editing is inheritably connected with a risk of off target effects. It would therefore be great to identify AML samples carrying the PPM1D mutation that has been naturally selected during the transformation process.

---

## [Author Response]

The following is the authors’ response to the original reviews.

We thank the reviewers for their insightful and constructive comments of our work that have helped to strengthen the manuscript. In response to the additional suggestions provided by the reviewers, we have made revisions by adding or replacing five main figures, three supplementary figures, refining the text, and clarifying certain conclusions. Detailed responses to the reviewers’ points can be found below.

Additional experiments, textual changes, or modulation of claims are needed to address weaknesses in the SOD1 portion of the study. Specifically:A) These studies require an assessment of the on-target efficacy of the inhibitors at the relevant concentration ranges. Ideally, they should have minimal effects against SOD1 knockout cell lines (an acute challenge at a time point before the growth defects become apparent) and show better efficacy in SOD1-overexpressing lines. Key experiments (changes in superoxide, OCR profiling, DNA alkaline comet assay) would be more convincing if they were carried out with SOD1 knockout lines to compare against the inhibitor effects (3-4 days after introducing sgSOD1 when growth defects are not apparent). In addition, SOD activity should be measured directly following inhibitor treatment.

We agree with the reviewers that the on- vs. off-target effects of the pharmacologic SOD1 inhibitors is a critical point to address. We have validated that SOD activity is reduced following treatment with ATN-224 in Figure 2 – Figure supplement 1A.

Nevertheless, we acknowledge that the potential for off-target effects of these inhibitors cannot be completely ruled out. To address this concern, we have incorporated a discussion regarding the potential off-target effects of both LCS-1 and ATN-224.

B) Assays should be included to support that SOD1 activity is altered. ATN-224 and LCS-1 are used to inhibit SOD1 function in the majority of the experiments, which should be supported by SOD activity assays to confirm SOD inhibition. Further, the concentration of ATN-224 used in this paper (12.5 uM) is beyond the concentration of what has been reported to inhibit SOD1 function in human blood cells. In Figure 4D, the authors demonstrate comparable SOD1 total protein levels in WT and PPM1Dmutant cells. However, the authors should further address whether PPM1D-mutation alters SOD1 activity via SOD activity assays.

We thank the reviewers for these suggestions. We have performed SOD activity assays which confirmed that SOD activity is inhibited upon treatment with ATN-224 at two concentrations (6.25 and 12.5 uM). Although we also did this for LCS-1-treated cells as well, in our hands, we did not see reduced SOD activity. However, LCS-1 has been shown to inhibit SOD activity in other publications including PMID: 21930909 and PMID: 32424294. From these assays, we have also found that PPM1D-mutant cells had increased SOD activity at baseline, despite having similar levels of SOD1 protein. These data have been added to Figure 2–Figure supplement 1A.

C) Some conclusions are not fully supported by the data provided. The authors claimed that "upon inhibition of SOD1, there was an increase in ROS that was specific to the mutant cells" in Figure 2E. Comparison of ROS levels among untreated, ATN-224, and LCS-1 of PPM1D-mutant cells should have been made and the statistics analysis among these groups should have been provided. Moreover, in Figure 2-Figure Supplement 1E, LCS-1 treatment does not increase ROS levels in PPM1D mutant LCLs. Performing these experiments with control and SOD1 deletion cells would have strengthened the results. Along with this point, the authors should comment on why SOD2 is not identified as a top hit in the CRISPR screen, as SOD2 deletion accumulates superoxide in cells.

After performing additional statistical analyses for Figure 2E, we found that the minor increase in ROS levels in the mutant cells after SOD1 inhibition was not statistically significant. We have revised the text accordingly.

As for why SOD2 was not identified as a top hit, we postulate that this may be due to inherent dependency of the WT cell lines on SOD2.

D) Fig. 1 - SOD1 appears to be clustered with several other genes in the volcano plot (including FANC proteins). Did any other ROS-detoxifying enzymes show similar fitness scores? The effects of the SOD1 sgRNA are striking, however, it would be useful to see qPCR or immunoblot data confirming robust depletion.

Thank you for your suggestion. We have validated the loss of SOD1 protein expression after SOD1 sgRNA deletion by immunoblot and have added this data to Figure 1– figure supplement 1D. While other ROS-detoxifying enzymes were not significantly enriched in the top 37 hits, interestingly, the Fanconi Anemia pathway also has roles in counteracting oxidative stress. FA-deficient cells have mitochondrial dysfunction and redox imbalance, and several of the FA family proteins are implicated in mitophagy. Therefore, there may be an interesting interplay between SOD1 and the FA pathway that is worth highlighting in the discussion of our manuscript even though there was no experimental investigation performed.

E) Fig. 2 - What are the relative SOD1 levels in the mutant PPM1D vs. WT. cell lines? The effects of the chemical inhibitors are stronger in MOLM-13 than in the other two lines. These data could also point to whether LCS-1 and ATN-224 cytotoxicity are on-target or off-target at these concentrations, which is a key issue not currently addressed in these studies. This is a particular concern as the OCI-AML2 line shows a stronger growth defect with CRISPR SOD1 KO (in Fig 1) but the smallest effects with these chemical inhibitors. The authors should also include SOD1 levels for Figure 1D and Figure 4Figure supplement 1C.

SOD1 protein expression is similar between WT and PPM1D-mutant cell lines and the loss of SOD1 after SOD1 sgRNA deletion was validated by immunoblot. These data have been added to Figure 1- figure supplement 1D and Figure 4D.

F) Does SOD1 co-expression in PPM1D-mutant patient AML correspond to poorer disease outcomes? This can be evaluated in publicly available patient datasets and would support the idea of SOD1 synthetic lethality.

Unfortunately, there are no publicly available patient datasets with sufficient cases of de novo PPMDmutant AML to assess this question.

G) While endogenous mitochondrial superoxide levels are elevated in PPM1D mutant lines, it is entirely unclear why SOD1 inhibition should affect mitochondrial superoxide as it detoxifies cytosolic superoxide. Also unclear why the DCFDA signal (which measures total hydroperoxides) is *increased* under SOD1 inhibition - SOD1 dismutates superoxide radicals into hydrogen peroxide, therefore unless SOD2 is compensating for SOD1 loss, one might expect hydroperoxides to be lower (unless some entirely different oxidase is increasing their levels). None of these outcomes appear to be considered. Finally, it is not explained how lipid peroxidation, which requires the production of hydroxyl or similarly high-potency radicals, is being caused by increased superoxide or peroxides. One possibility is there is an increase in labile iron, in which case this phenotype would be rescued by the iron chelator desferal, and by the lipophilic antioxidant, ferrostatin.

We measured intracellular labile iron levels by flow cytometry by staining the cells with FerroOrange at baseline and after SOD1 inhibition with our pharmacologic inhibitors (ATN-224 at 12.5 uM and LCS-1 at 1.25 uM). Across the three leukemia cell lines, we saw variable results in iron levels with no appreciable patterns (see below). Therefore, we cannot make conclusions about the contribution of labile iron to our observed phenotypes.

**Author response image 1. sa4fig1:** 

H) Do the sgSOD1 cells also show similar increases in MitoSox green, DCFDA, and BODIPY signal? These experiments would clarify whether the effects of the inhibitors are directly related directly to SOD1 loss or if they represent off-target effects from the inhibitors and/or compensatory changes in SOD2.

We do not observe changes in SOD2 in the several contexts in which we have examined this. We cannot exclude off-target effects of the inhibitors so have clarified this in the text.

I) The authors may want to assess whether Rac1 or NADPH oxidase activity is altered in the SOD1 KO in WT vs. PPM1D cells. Their results may be the consequence of compromised ROS-driven survival signaling or DNA repair rather than direct ROS-induced damage, which is not caused directly by superoxide (or hydrogen peroxide).

We appreciate the reviewer’s recommendations. However, due to time constraints, we regret not being able to assess Rac1 or NADPH oxidase activity. Nevertheless, we recognize the possibility of altered ROS-driven signaling rather than ROS-induced damage as a driver of our phenotype and have incorporated this possibility into our discussion.

J) Fig. 3 - the effects on mitochondrial respiratory parameters, while statistically significant, do not seem biologically striking. Also, these data are shown for OCI-AML2 cells which show the smallest cytotoxic effects with the SOD1 inhibitors among the 3 lines tested. They do however show the most robust growth defect with sgSOD1. This discrepancy could suggest that mitochondrial dysfunction does not underlie the observed growth defect and/or the inhibitor cytotoxicity is not on-target. Ideally, mitochondrial profiling should also be carried out on this cell line with inducible SOD1 depletion. Have the authors assessed whether the mitochondrial Bcl family proteins are affected by the inhibitors?

We assessed a few members of the mitochondrial Bcl-family proteins including MCL-1, BCL-2, and BCL-XL during the revision process. PPM1D-mutant cells have mildly increased expression of these anti-apoptotic proteins at baseline and the expression is not altered by pharmacologic SOD1 inhibition (see Author response image 2 below). Due to time constraints, we were unable to perform seahorse assays and mitochondrial profiling in the SOD1-deletion cells.

**Author response image 2. sa4fig2:** 

K) Fig. 4 - Currently the data in this figure do not support the authors' claim that PPM1D-mutant cells have impaired antioxidant defense mechanisms, leading to an elevation in ROS levels and reliance on SOD1 for protection. It should be noted that oxidative stress specifically refers to adverse cellular effects of increasing ROS, not baseline levels of various redox parameters. Ideally, levels of GSSG/GSH would be a better measure of potential redox stress tolerance than the total antioxidant capacity assay. Finally, oxidative stress can be assessed by challenging the wt and mutant PPM1D cell lines with oxidant stressors such as paraquat which elevates superoxide, or drugs like erastin which elevate mitochondrial ROS. The immunoblot shows negligible changes in the antioxidant proteins assayed. Again, this blot should include SOD2 which is the most relevant antioxidant in the context of mitochondrial superoxide.

We measured intracellular glutathione levels by flow cytometry and found that PPM1D-mutant cells had a greater proportion of cells with low levels of GSH. This data has been added as Figure 4D. We have also repeated the western blot to look at the antioxidant proteins catalase, SOD1, and thioredoxin after SOD1-deletion and pharmacologic SOD1 inhibition. We evaluated SOD2 protein levels in these experiments, as suggested. Smooth muscle actin (SMA) is included in the antibody cocktail as a loading control. However, it is unclear to us as to why PPM1D-mutant cells consistently have significantly higher levels of SMA. Therefore, we included a separate loading control, Vinculin. Repeat of these western blots showed a clearer difference between WT and PPM1D-mutant cells in the levels of these antioxidant proteins in which PPM1D-mutant cells have decreased levels of catalase and thioredoxin. These blots also show that SOD2 levels may be mildly increased in the PPM1D-mutant cells at baseline but is not significantly upregulated upon SOD1 inhibition. We have replaced the original immunoblot from Figure 4D with the revised blots that more clearly demonstrate the reduced levels of catalase and thioredoxin, now figure 4E.

L) Fig. 5 - These data support that DNA breaks are elevated in PPM1D mutant vs. wt cells. However, the data with the chemical SOD1 inhibitor again do not convince us that the enhanced levels are due to on-target effects on SOD1. Use of the alkaline comet assay is appropriate for these studies and the 8-oxoguanine data do indicate contributions from oxidative DNA base damage. But these are unlikely to result directly from altered superoxide levels, as this species cannot directly oxidize DNA bases or cause DNA strand breaks.

Thank you to the reviewers for raising this point. We have performed comet assays in SOD1-deletion cells to look at levels of DNA damage. Consistent with the reviewers’ point, we do not see a significant increase in DNA breaks after SOD1 deletion. We have removed the data using the SOD1 inhibitor and instead show the COMET analysis in the PPM1D-mut and SOD1-KO cells (see Figure 5F). We now make the point that increased DNA damage with SOD1 loss cannot explain the vulnerability of the double-mutant cells.

M) Instead of using NAC, which elevates glutathione synthesis but also has several known side effects, the authors may want to determine whether Tempol, a SOD mimetic can rescue the effects of SOD1 knockout or inhibition. This would directly prove that SOD1 functional loss underlies the observed growth defect and cytotoxicity from genetic SOD1 knockdown or chemical inhibition.

This is an excellent suggestion; we have added comments to this effect into the discussion.

N) It is recommended the discussion focus more strongly on how the signaling function of superoxide vs. its reactions with other molecular entities to induce genotoxic outcomes could be contributing to the observed phenotypes. The discussion of FANC proteins, which were targets with similar fitness scores but not experimentally investigated at all, is an unwarranted digression.

Thank you for this recommendation. We have expanded the discussion to focus more on the signaling functions of superoxide. However, considering the role of the Fanconi Anemia pathway in mitigating DNA damage and oxidative stress, we believe the discussion on the FANC proteins is important due to the possible intersection with SOD1. Therefore, we have refined this portion discussion to focus more on the interplay between SOD1 and FA.

O) The complete lack of consideration of SOD2 in these studies is a missed opportunity as it reduces mitochondrial superoxide levels but elevates hydrogen peroxide levels. It would be very interesting to see whether SOD1 inhibition leads to compensatory increases in SOD2. SOD2 can be easily measured by immunoblot. Furthermore, measuring total superoxide via hydroethidium in a flow cytometric assay vs. mitochondrial ROS in PPM1D mut vs. wt cells and under SOD1 knockout would enable a determination of which species dominates (cytosolic or mitochondrial). These experiments are required to fill some logical gaps in the interpretation of their redox data.

During the revision process, we have included SOD2 in our studies and have found that loss of SOD1 via genetic deletion and pharmacologic inhibition does not lead to compensatory increases in SOD2 (Figure 4D). Additionally, we have measured cytoplasmic superoxide levels using dihydroethidium to differentiate between cytoplasmic vs. mitochondrial superoxide. We found that at baseline levels, the mutant cells also harbored more cytoplasmic superoxide. We have added this figure as Figure 2C and moved the original mitochondrial superoxide data to Figure 2-figure supplement 1C.

P) Given the DNA breaks observed in PPM1D mutant cells, it is highly recommended that the authors assess whether iron levels are elevated in mut vs. wt cells and whether desferal can rescue observed SOD1 inhibition defects. Also, it has been reported that PPM1D promotes homologous recombination by forming a stable complex with BRCA1-BARD1, thereby enhancing their recruitment to doublestrand break sites. The authors should comment on why there is no difference in repair via HR in WT and PPM1D mutant cells in Figure 5C.

Please see comment G regarding our findings about iron levels.

The reviewers pose an interesting question as to why there is no difference in HR repair between WT and mutant cells, given the reported role of PPM1D in promoting HR. We have addressed this question in the main text. We believe that several factors can limit the extent of HR enhancement in PPM1D-mutant cells. For example, HR is typically confined to the S/G2 phase and thus may be constrained by cell cycling, among other regulatory mechanisms.

Other comments:A) The authors described in the Method section that "The CRISPR Screen PPM1D mutant Cas9expressing OCI-AML2 cell lines were transduced with lentivirus library supernatant." The authors need to provide information on whether the MOI of the CRISPR screen has been well controlled to ensure that the majority of the cell population has a single copy of sgRNA transduction.

We performed a lentiviral titer curve prior to the screen to determine the volume of viral supernatant to add for a multiplicity of infection (MOI) of 0.3. This important detail has been added to our Methods.

B) The study convincingly shows differences between parental leukemic cells and the PPM1D mutants but one important control is missing in experiments related to Fig. 2 and 3. All PPM1D mutant clones used in this study were subjected to the blasticidin selection of the transduced cells to generate cells stably expressing Cas9 and subsequently, the clones with successful PPM1D targeting were expanded. The authors should demonstrate that increased ROS production is not just a consequence of the lentiviral transduction and antibiotic selection and that it corresponds to increased PPM1D activity in PPM1D mutant cells. To do that, authors could compare PPM1D clones to parental cells that underwent the same selection procedure (OCI-AML2-Cas9 cells and OCI-AML3-Cas9 cells).

It is true that the parental OCI-AML2 and OCI-AML3 cell lines underwent four days of blasticidin selection to create the stably expressing Cas9 cell lines. However, after the four-day period, the blasticidin was removed from the cell culture media. From there, we induced the PPM1D-mutations into the Cas9-expressing “WT” cell lines using the RNP-based CRISPR/Cas9 delivery method and single cells were then sorted into 96-well plates. Clones were expanded and validated using Sanger sequencing, TIDE analysis, and western blot. In all of our assays, we compare the WT Cas9 cells to the PPM1D-mutant Cas9 cells. Additionally, the cells have been expanded and passaged several times after blasticidin-selection. Therefore, we believe it is unlikely that there are residual ROSinducing effects from the antibiotic treatment.

C) The authors mention that they identified 3530 genes differentially expressed in parental and PPM1D mutant cells (line 267) but it is unclear what was the threshold for statistical significance. They mention FDR<0.05 in the Methods but show GSEA analysis with FDR<0.25 in Figure 4A. Source data for Fig. 4 is missing and the list of differentially expressed genes is not shown.

The source data files for Figures 1 and 4 will be uploaded with the revised manuscript. Upon reviewing the source data, we noticed an error in the number of differentially expressed genes. We have corrected this in line 274 and you will see that this correlates with Figure 4-source data 1. For the thresholds, we used an FDR<0.05 for the differential gene expression analysis, and an FDR <0.25 in the GSEA, which is an appropriate threshold for GSEA. We have clarified these thresholds in the methods section.

D) Include a definition of MFI in Figure legend Fig.2 and also in the Methods section. The unit should be indicated at both the x and y axes.

We have defined MFI in the figure legends and methods sections and have updated the figures accordingly.

E) Legend to Figure 2 - Figure Supplement 1 E should define the grey and pink columns (likely WT and mutants LCLs).

Thank you. We have defined the grey and pink columns as WT and PPM1D-mutant cell lines, respectively for Figure 2 – Figure supplement 2D and E.

F) Reporter assays in Fig. 5 convincingly show that NHEJ capacity is reduced in PPM1D mut cells. In the text, the authors state that this might reflect the impact of PPM1D on LSD1 (line 365). Although this might be the case, other options are equally possible. It would be appropriate to include a reference to the ability of PPM1D to counteract gH2AX and ATM which generate the most upstream signals in DDR.

Thank you to the reviewers for raising this excellent point. We have revised the text to incorporate the impact of PPM1D on yH2AX and ATM on NHEJ.

G) The authors correctly state that truncation of PPM1D leads to protein stabilization (line 85) and that it is present in U2OS cells (line 355). These observations have first been reported by Kleiblova et al 2013 and therefore one reviewer believes that this reference should be included. This study also identified truncating PPM1D mutation in colon adenocarcinoma. HCT116 cells and the role of PPM1D mutation in promoting the growth of colon cancer has subsequently been tested in an animal model (Burocziova et al., 2019).

Thank you. We have added this reference to our text in line 360.